# A genome-wide comprehensive analysis of nucleosome positioning in yeast

**Leo Zeitler** [ORCID], **Kévin André, Adriana Alberti, Cyril Denby Wilkes** [ORCID]\*, **Julie Soutourina** [ORCID]\*, **Arach Goldar** [ORCID]\*

Université Paris-Saclay, CEA, CNRS, Institute for Integrative Biology of the Cell (I2BC),Gif-sur-Yvette, France

\* cyril.denby-wilkes@cea.fr (CDW); julie.soutourina@cea.fr (JS); arach.goldar@cea.fr (AG)

**Data Availability Statement:** During this study, no new data was generated. Codes are made available on Zenodo (DOI 10.5281/zenodo.8335111) Processed data are available at Figshare (https://doi.org/10.6084/m9.figshare.25011752.v1). As

## Abstract

In eukaryotic cells, the one-dimensional DNA molecules need to be tightly packaged into the spatially constraining nucleus. Folding is achieved on its lowest level by wrapping the DNA around nucleosomes. Their arrangement regulates other nuclear processes, such as transcription and DNA repair. Despite strong efforts to study nucleosome positioning using Next Generation Sequencing (NGS) data, the mechanism of their collective arrangement along the gene body remains poorly understood. Here, we classify nucleosome distributions of protein-coding genes in *Saccharomyces cerevisiae* according to their profile similarity and analyse their differences using functional Principal Component Analysis. By decomposing the NGS signals into their main descriptive functions, we compared wild type and chromatin remodeler-deficient strains, keeping position-specific details preserved whilst considering the nucleosome arrangement as a whole. A correlation analysis with other genomic properties, such as gene size and length of the upstream Nucleosome Depleted Region (NDR), identified key factors that influence the nucleosome distribution. We reveal that the RSC chromatin remodeler—which is responsible for NDR maintenance—is indispensable for decoupling nucleosome arrangement within the gene from positioning outside, which interfere in *rsc8*-depleted conditions. Moreover, nucleosome profiles in *chd1Δ* strains displayed a clear correlation with RNA polymerase II presence, whereas wild type cells did not indicate a noticeable interdependence. We propose that RSC is pivotal for global nucleosome organisation, whilst Chd1 plays a key role for maintaining local arrangement.

## Author summary

In baker's yeast, as in other living organisms, the support of genetic information is tightly packaged and separated from the rest of the cell. Folding is achieved on its lowest level by wrapping the DNA around molecular complexes called nucleosomes. However, the survival of the baker's yeast requires that essential cellular processes access the genetic information protected by nucleosomes. Therefore, the nucleosome arrangement along the genome should modulate the access and use of this information and ultimately the functioning of the cell. Despite strong efforts to study nucleosome profiles, the mechanism of their collective arrangement along the genome remains poorly understood. Here, we

described in Methods section all raw data can be accessed at GEO accession numbers GSE69400 and GSE73428.

**Funding:** This work was supported by Fondation ARC [PGA1 RF20170205342]; Comité Ile-de-France - La Ligue Nationale Contre le Cancer. K.A. was supported by a PhD training contact from the French Ministry of Higher Education and Research. L.Z. was supported by a PhD training contract from the CEA NUMERICS program, which has received funding from European Union's Horizon 2020 research and innovation program under the Marie Sklodowska-Curie grant agreement No 800945. This project has received financial support from the CNRS through the MITI interdisciplinary programs. The funders had no role in study design, data collection and analysis, decision to publish, or preparation of the manuscript.

**Competing interests:** The authors have declared that no competing interests exist.

tackled this issue by comparing and classifying directly the nucleosome profiles along the genes (genomic regions that are known to support the necessary informations for cell functioning). This drives us to highlight that in baker's yeast the spatial organisation of nucleosomes in genes is different from other genomic regions and this difference is maintained actively by energy consuming factors. We show that the regulation and compartmentalisation of nucleosomal organisation require the concomitant actions of local and global processes.

## Introduction

The eukaryotic DNA must be tightly wrapped into the spatially constraining nucleus. This is achieved in the form of chromatin, a DNA-protein complex within which the 1-dimensional DNA is condensed around histone octamers and folded to a 3-dimensional structure. To be more precise, these histone complexes are positively-charged multiprotein structures around which the DNA molecule is locally coiled, forming a linear organisation resembling the stringing together of beads. This is why the primary structure of chromatin is commonly represented by a so-called *beads-on-a-string* model. In yeast, a nucleosome refers to ≈147 base pairs (bp) of DNA that are wrapped around four histone units. Nucleosomes are closely spaced, with an averaged centre-to-centre distance of 165 bp, leaving roughly 15 bp of linker DNA between two adjacent histone complexes. There is a consensus that phasing is highly regular within coding regions, which is interrupted by Nucleosome Depleted Regions (NDRs) between two neighbouring genes. This observation gave rise to the barrier model, which proposes that promoter-dependent properties (e.g. bound proteins or sequence composition) pose a limit for nucleosome assembly, and arrangement occurs with respect to this barrier [1, 2]. However, it is widely accepted that various factors establish and influence the genome-wide positional nucleosome landscape, including sequence composition, transcription, and chromatin remodelers [3–6]. Since the DNA molecule must bend to wrap around the histone octamer, the local nucleotide sequence naturally affects positioning. Generally speaking, GC-rich sequences are more flexible than AT-rich ones, and they are favorable to support the presence of a nucleosome [7, 8]. However, sequence-related properties might be dependent on specific motifs.

The condensed packaging also functions as regulator for various DNA-protein interactions. Most of these processes rely on chromatin remodeler complexes, which can—by consuming energy obtained from ATP hydrolysis—move, add, or evict the histone complexes to provide or inhibit direct access to the DNA sequence [9]. In yeast, chromatin organisation is maintained by four protein families, SWI/SNF, INO80, ISW, and CHD. The RSC remodeler complex of the SWI/SNF family is the only essential chromatin remodeler in *Saccharomyces cerevisiae*, and it is recruited to promoter regions where it is responsible for the maintenance of NDRs [10–12]. It has also been reported that the complex has an influence on nucleosome organisation in coding regions as well as supporting RNA Polymerase II (Pol II) elongation [13]. It is presumed to restore chromatin organisation after transcription [14]. However, RSC does not exhibit an impact on regular nucleosome spacing within the gene [14, 15]. Chd1—the only member of the CHD remodeler family in yeast—is associated with various transcription-regulating functions, including initiation, elongation, and termination [16]. It has been suggested that Chd1 stabilises perturbed nucleosomes during gene expression [17]. Isw1 and Chd1 are supposed to antagonise for nucleosome spacing within the gene, with Isw1 dominating profiles along genes with larger spacing, whereas Chd1 seems to control shorter spacing [12, 18]. It has been reported that deletion of Chd1 and Isw1 only disrupt inter-nucleosome

distances and leave the +1 position unaffected [19]. Isw2 is similarly associated with regular spacing [20], and it is particularly affecting nucleosomes close to the NDR, which is presumed to regulate transcription [21]. However, the underlying mechanism for chromatin remodeling is still under debate, and a scientific consensus is missing [22–25].

Several studies showed an interdependence between nucleosome distribution and gene expression by using MNase-seq data, a Next Generation Sequencing (NGS) technique that allows the measurement of nucleosome profiles by using MNase digestion of purified chromatin [26, 27]. It has been suggested that high gene expression correlates with low nucleosome regularity [28] as well as extreme spacing (both short and long) [18]. There are contradicting results about the correlation between transcription and nucleosome phasing. Whilst [18, 29], and [30] report that transcription increases random positioning and weakened phasing, [28] show that nucleosome phasing of highly expressed genes is increased. The depletion of Pol II exhibited increased array regularity [31]. This phenomenon seems to be conserved across species, as indicated by studies using *Drosophila* [28] and mouse cell lines [32]. The outcomes indicate that gene expression can be partially explained by nucleosome positioning over the gene body. Nonetheless, the autocorrelation of MNase-seq profiles along genes revealed that nucleosomal organisation accounts for only ≈25% of the observed transcriptional variability, even though genes with similar regularity tend to have the same level of gene expression [33]. Surprisingly, many strains deficient for chromatin remodelers seem to show only a marginal effect on transcription [18, 19]. The only exception is *rsc8*-depleted cells, which exhibit a global decrease in gene expression [12]. A clear picture between nucleosome phasing and Pol II presence is still lacking.

Different approaches have been used to categorise collective nucleosome arrangement within transcribed regions using NGS data. However, many of them rely predominantly on measurements that describe only an average over the entire profile, such as autocorrelation measurements [33] or Pearson correlation that was adapted to include coverage [34]. Pearson correlation was also used to compare nucleosome positioning of genes before and after replication [35]. Another analysis that takes into account multiple nucleosomes upstream and downstream of the NDR was presented by [14]. However, the study focused on changes with respect to the NDR, and many phenomenological descriptions are based on the application of different analysis techniques. In order to provide comparability of nucleosome positioning changes between various mutants, we aimed to use a single mathematical framework that can be applied to all strains. To our knowledge, a unifying approach assessing location-specific phasing properties along the entire nucleosome array over varying conditions has not been proposed, and a direct comparison of the effects in different remodeler-deficient strains is difficult.

In this work, we present a genome-wide analysis of collective nucleosome positioning along the gene. We define nucleosome positioning and phasing to be the positions of the MNase-seq signal peaks over an entire single nucleosome array. By clustering the MNase-seq signals of coding regions along 6–7 histone complexes into two groups using linear Pearson cross-correlation—which measures similarity of the entire nucleosome arrangement between each gene pair—we can categorise coding regions according to their likely phasing similarity imposed by chromatin remodelers. In order to interpret how profiles are classified into the two groups, we combined the clustering with an alternative data representation via functional Principal Component Analysis (fPCA). Whilst related to the conventional Principal Component Analysis (PCA), it assumes a functional relationship between positions along the profile, whereas PCA conjectures independence of every base pair along the gene. Therefore, fPCA implicitly considers spatial dependency, which is a fundamental assumption in common nucleosome phasing models like the barrier model, where nucleosomes phasing is coordinated with respect to a

barrier and each other. FPCA is commonly used in time series and signal processing, and it has been used in biology for analysing crop yield [36], identifying child growth patterns [37], as well as studying genetic variation and the allelic spectrum [38]. However, it has never been applied to the spatial interdependence of nucleosome phasing to our knowledge.

The established Pearson clusters can be visually separated by considering only two fPCs, which are therefore sufficient to interpret the gene groups. Using our analysis, we can repeatedly investigate nucleosome distributions of different chromatin remodeler-mutant strains using the same framework and interpret major differences along the entire nucleosome arrangement. By relating Pearson correlation with spatial properties along the profiles, our approach refines and complements other studies that focused either on a few individual nucleosomes close to the NDR or Transcription Starting Site (TSS); or which assessed only the average correlation of the entire array (e.g. via autocorrelation). Using MNase-seq data from yeast strains deficient for different chromatin remodelers [12, 18], we reveal that Rsc8 strongly limits coordinated nucleosome arrangement to the transcribed region. It might be therefore responsible for gene-specific phasing. By measuring how the Pearson cluster separation changes between mutants using a Support Vector Machine (SVM), we identified 5 combinations of gene deletions or protein depletions which have a notable impact on phasing properties along the entire nucleosome array compared to Wild Type (WT) conditions. Measuring correlation with other nuclear processes disclosed that none of the commonly assumed factors can easily explain long-reaching nucleosome arrangement in WT strains within the gene body. However, gene deletions—in particularly mutants that contained *chd1Δ*—caused a strong correlation with Pol II presence. Our results indicate a new mechanistic understanding of chromatin remodelers, where Rsc8 is responsible for long-range coordination and Chd1 for local positioning of nucleosomes. All customised source code was made available on Zenodo (DOI 10.5281/zenodo.8335111).

## Results

### Nucleosome profiles can be well distinguished based on their coordinated positioning in WT

In order to compare nucleosome profiles over the gene body in WT conditions, we measured the pairwise Pearson cross-correlation of the MNase-seq data produced by [12, 18] for all protein-coding regions [39] using Eq 1. The Pearson correlation index is positive when the sequencing signals of both genes tend to change towards the same direction at the same position; and it is negative when one profile is likely to increase whereas the other one decreases. Therefore, it compares similarity of the distributional shape—i.e. whether genes are apt to contain nucleosomes at similar positions—and it does not take the scaling of the sequencing data into account. The entire arrangement for each gene is treated as an entity. For both replicates, we considered 1000 bp after and 200 bp before the +1 position (= 1200 bp, approximately the average size of a gene in *Saccharomyces cerevisiae*), containing 6–7 nucleosome dyads.

These Pearson coefficients were used as a distance metric to cluster nucleosome profiles into distinct partitions using *k*-mean clustering. In a nutshell, the algorithm divides a data set of *m* observations (the MNase-seq data) into *k* groups by minimising the variance within each cluster based on a distance metric (here, the pairwise Pearson indices over all genes). Therefore, genes within a group tend to have nucleosomes at comparable positions, whereas profiles of different groups are likely to be less similar. Using a silhouette criterion measurement—which compares the similarity of an object to its own cluster with the similarity to other clusters—we determined that the Pearson coefficients are most distinctly divided when *k* = 2 (i.e. when having two groups, Fig 1A). As the *k*-mean clustering algorithm aggregates profiles

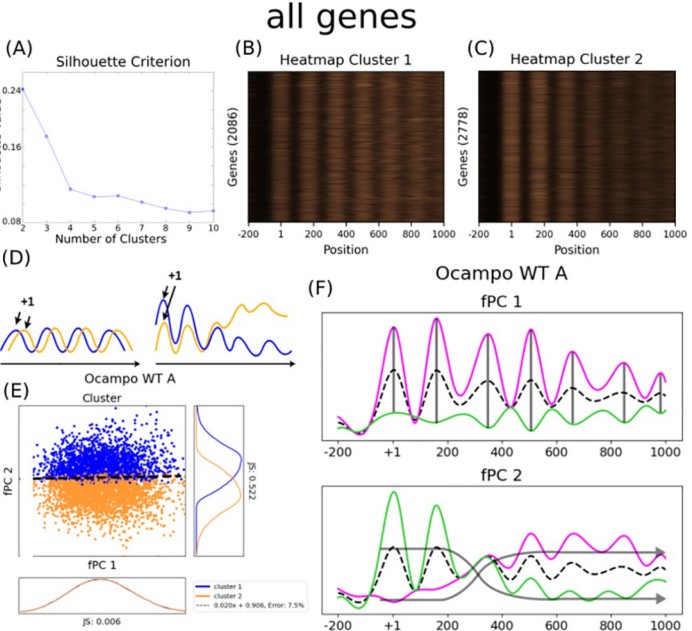

**Fig 1. Pearson clusters and fPCA considering all protein-coding genes.** (A) The silhouette plot clearly indicates that the data can be best divided into two clusters, and creating more groups would only decrease the difference between each cluster. (B) and (C) display the profiles for each cluster. Large values are given in copper, low values are black, and the colour gradient in between is uniformly scaled. It is therefore a perceptually uniform representation. Both heatmaps are normalised independently, such that their respective largest value is displayed in the strongest copper hue and their lowest value in black. Unfortunately, it is difficult to quantify visually why these clusters were established. This is particularly true because the Pearson index measures only general trends in the profile, and it does not take the scaling into account. Each row represents a gene, and the x-axis shows the position along the coding region, with the +1 nucleosome defined to be at position 0 bp. The colour code represents MNase-seq amplitude, i.e. copper values show large MNase-seq signal values, whereas dark areas indicate a low amplitude. (D) The cartoon presents the hypothesised differences that could occur between the Pearson clusters. Due to the well-positioned nucleosomes and the wave-like structure of MNase-seq data, we presume that the Pearson correlation measures coordinated nucleosome positioning along the gene. If two profiles (orange and blue) are in two different clusters, this could indicate either a shift in the exact nucleosome positions (left); or a general trend in the MNase-seq signal amplitude, i.e. either increasing or decreasing (right). (E) Pearson clusters considering all genes are linearly separable with respect to their fPC scores. This indicates that two fPCs are sufficient to interpret the gene groups. We use the symmetric Jensen-Shannon (JS) distance to describe separability between the clusters along fPC1 and fPC2. The JS distance between the cluster distributions is much larger for fPC 2 than for fPC 1. Orange and blue indicate each one group, the dashed line symbolises the best linear separation using a SVM. The x-axis represents the score of the first fPC $\zeta^1$, the y-axis gives the score for the second fPC $\zeta^2$. Both axes are scaled to the same range, points outside the range (29) were included in the analysis but not plotted. (F) When analysing the effect of the major fPCs, they describe predominantly position-dependent scaling (transparent black lines, fPC 1) and collective nucleosome phasing (transparent black arrows, fPC 2). The second fPC in WT indicates an increasing or decreasing signal magnitude as a function of distance from the TSS, suggesting stronger or weaker presence (corresponding to panel D right). The mean is given as a dashed black line, a positive contribution—i.e. adding the fPC to the mean—is displayed in magenta, and a negative contribution—i.e. subtracting from the mean—is shown in green. Trends over the entire array are indicated by grey arrows. When exact positions were seemingly not affected by the fPC, we marked the positions with a grey vertical bar. See Methods for more information about how the plots were produced.

together with a large pairwise Pearson index, we expected that the distribution over all pairwise Pearson correlations between nucleosome arrangements from different clusters is minimal (in the following called inter-cluster correlation). We validated the significance of the obtained Pearson gene groups by creating 500 random clusters and calculating their p-value using a one-sided Kolmogorov–Smirnov (KS) test. We averaged the p-value over the KS test result over all 500 repetitions to smooth out random fluctuations (average p-value 0.0009, S1 Fig).

This shows that nucleosomal arrays can be significantly separated into two groups using linear correlation of MNase-seq data between genes (Fig 1B and 1C).

It is difficult to straightforwardly determine how the *k*-mean clustering algorithm distinguishes between these two groups; yet the interpretation of the discriminating boundary could reveal important insights about the nucleosome positioning that is presumably imposed by chromatin remodelers. As the data by Ocampo et al. [12, 18] contains several mutants, we want to identify this discriminator repeatedly with the same mathematical framework to make the results comparable. Due to the nature of the Pearson correlation index, we can make the following assumptions. As nucleosomes are commonly well positioned in budding yeast, the MNase-seq data resembles a wave-like function with one peak approximately every 200 bp. Moreover, single histone complexes cannot overlap in a single cell. The Pearson correlation measures therefore the average phasing similarity of the entire nucleosome array of two genes. Differences in similarity come either from shifts in exact positioning (i.e. well-defined peaks, Fig 1D left) or from a change in the signal amplitude (i.e. increasing or decreasing MNase-seq magnitude over the profile or at particular locations, Fig 1D right). The clusters must be separated based on either of these two trends, or possibly a combination of them.

In the following, we refer with *coordinated positioning* to the configuration of the entire nucleosome array, and consequently, to their behaviour with respect to the two separating trends of the *k*-mean clustering. Unfortunately, the Pearson coefficient measures only the average linear pairwise correlation over the entire profile, rather than taking position-dependent particularities into account. Therefore, simply extracting the boundary from the *k*-mean clusters does not explain whether the groups were established with respect to a shift or a change in amplitude (i.e. the previously determined discriminators). Instead, it is possible to investigate how the clusters distribute with respect to the data itself; or, alternatively, with respect to a different description using dimensionality reduction methods. By evaluating the major differences between the two groups of genes, we can interpret the separating clustering boundary and link it to particular properties along the nucleosome profile.

Conventional approaches apply dimensionality reductions like PCA to visually analyse clustering distributions. However, using PCA would implicitly mean that we assume independence between every position along the gene. By using the Pearson correlation measurement, we treat every profile as a single entity, which would be violated by the independence conjecture. This also contradicts the fundamental assumption of the barrier model where the positioning of earlier nucleosomes affect later phasing. Instead, we understand the arrangement as the result of a coordinated process. We assume that the MNase-seq signal along each gene can be described as a single (unknown) continuous function, which can be approximated by a mixture of a finite number of known simpler functions (so-called basis functions). In this study, we used 20 B-Splines to represent the MNase-seq data along each gene, which were subsequently averaged to a mean profile. This permitted the application of fPCA to determine the two best-characterising functional Principal Components (fPCs) that describe each nucleosome arrangement. It incorporates specific assumptions about the spatial relationship in the distribution through the basis functions, which is the crucial difference between conventional PCA and fPCA. To be more precise, the establishment of the MNase-seq distribution is understood as a stochastic process with a mean behaviour. Each considered nucleosome array can be regarded as a realisation of this stochastic process with a deviance from the expected average distribution. Instead of defining a data representation for every gene individually, fPCA determines how the mean profile needs to be transformed to approximate a particular gene. This transformation is found by combining the basis functions over all coding regions to more complex functions that are orthonormal to each other and describe the most variance along the data (i.e. the fPCs, Eq 4). These functions transform the mean by adding them to the

average profile with a gene-specific scaling factor (i.e. $\zeta_i^j$ for the $j$-th fPC of the $i$-th gene). Consequently, every nucleosome array can be also described exclusively by the factors $\zeta_i^j$ together with the respective fPCs, and we can evaluate how the two Pearson clusters distribute with respect to these factors.

Interestingly, the two clusters—which were independently obtained by classical hierarchical $k$-mean clustering of Pearson coefficients—are visually neatly separated by using only the first two fPCs, indicating that they are sufficient to quantify the difference between the two sets of genes (Fig 1E). In fact, the separating boundary is almost exclusively dependent on the second fPC, whilst it is seemingly independent of the first. This is slightly less clear for the $B$ replicate, although still distinct (S2(B) Fig). Using our previous considerations about how the algorithm establishes the two clusters, we intuited that the second fPC describes coordinated nucleosome phasing along the gene body. By analysing the effect of the second fPC on the function shape, we conclude that the clusters are determined based on the downstream presence of nucleosomes (corresponding to the right cartoon in Fig 1D). We found that the first fPC largely represents amplitude scaling at a given position, as it does not influence the location of the peak (Fig 1F). The analysis shows that position-dependent amplitude scaling and coordinated arrangement are the best two independent functional descriptors for the MNase-seq data. We show and discuss the effect of the 10 major fPCs that describe most variance in S3 Fig. Despite the fact that the ratio of explained variance is not high (21.4% and 11.5% for fPC1 and fPC2, respectively), they are completely sufficient to distinguish between the Pearson correlation groups and permit an interpretation of the linear separating boundary between the clusters.

## FPCA reveals size-dependent Rsc8-mediated phasing of nucleosome positions

Since the smallest genes are ≈300 bp long, the 1000 bp window after the +1 position can contain much more than the actual length of the coding region. In order to analyse how nucleosome phasing is affected by the gene size, we repeated the fPCA considering exclusively small ($\leq$ 1000 bp, ≈26.7%) or large genes (>1000 bp, ≈73.3%). Consequently, the mean as well as the two fPCs changed, whilst we kept their allocation to the previously determined Pearson clusters the same (in the following also referred to as *all-gene* clusters). If coordinated positioning is substantially affected by the length of the transcribed region, we expected that the factors $\zeta_i^j$ of the two major fPCs should exhibit a changed behaviour with respect to the linear separability. We can confirm that the linear separation is preserved for large genes, although the boundary becomes slightly sloped (S2(C) and S2(D) Fig). The fPCs for only large genes are almost identical to the *all-gene* fPCs (S5 Fig). We therefore presume that the clusters can be still largely separated by the second fPC. We also considered a possible impact of the downstream NDR by analysing exclusively very large genes ($\geq$ 3000 bp, ≈11.5%). Once again, the boundary was clearly visible (S2(G) and S2(H) Fig). We concluded that the MNase-seq distribution over the first 6–7 nucleosomes of all genes larger than 1000 bp can be best clustered by the collective positioning, and it can be surmised that phasing within the gene body is only negligibly affected by the downstream NDR or nucleosomes outside the 1000 bp window.

However, the neat separation between the two clusters fully vanished for small genes (Fig 2A, for replicate B S2(E) Fig). Almost all data points belong to the same group, although both are present. We want to remind that clusters were established using all coding regions, whereas the functional representation depends now exclusively on genes smaller than 1000 bp. The newly determined fPCs include overlapping positioning inside and outside the gene body due to their varying size (S4 Fig). The fact that the clusters are not separable indicates that coordinated nucleosome phasing disappears after the Transcription Termination Site (TTS), and we

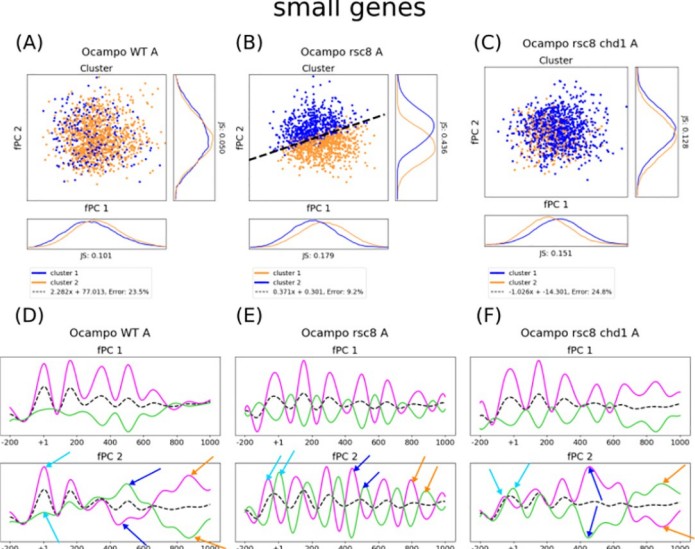

**Fig 2. Nucleosome phasing is strictly limited to the gene body, which is maintained by Rsc8 but antagonised by Chd1.** The cluster distribution plots in panels A-C show the distribution of both gene groups with respect to the small-gene fPCs of WT, *rsc8*-depleted cells, and *chd1Δ* strains. Orange and blue indicate the two clusters, and the black dashed line shows the separating boundary determined by a linear SVM. The histograms present the cluster distribution with respect to each axis. Panels D-F display the transformation of the average small-gene nucleosome profile by the two major fPCs for WT, *rsc8* depletion, and *chd1Δ*, respectively. The dashed black line as well as the solid lines in magenta and green display the mean, a positive contribution of the fPC, and a negative contribution. Turquoise arrows indicate the effect on the +1, dark blue arrows on the +4, and orange arrows on the +6 position. (A) When plotting the cluster distribution with respect to small-gene fCPs in WT, the linear separability is lost. (B) The fPCs of the *rsc8*-depleted strain maintain the linear separability, despite the fact that the groups were established for all genes. As we interpret the Pearson clusters as similarity in positioning between genes of 1000 bp mediated by chromatin remodelers, it possibly suggests that positioning outside coding regions influences nucleosomes inside and vice versa. (C) Whilst most mutants that were *rsc8* depleted could discriminate between the all-gene clusters using small-gene fPCs, this separability is lost again in *rsc8*-depleted *chd1Δ*, revealing partly antagonistic roles to maintain gene-specific phasing for Rsc8 and Chd1. (D) The effect of two fPCs sheds light on why the Pearson groups are not linearly separable in WT using small-gene fPCs. The distribution of the second fPC changes its regular wave-like form to much broader peaks and valleys after the +2 nucleosome, which corresponds to approximately the size of the smallest genes in budding yeast. (E) Nucleosome positioning in *rsc8*-depleted conditions is clearly visible along the entire considered region, despite the included genes being smaller. This suggests that gene-specific nucleosome arrangement cannot be maintained. It is of note that the phasing also changes for the +1 nucleosome, and the NDR can be seemingly not conserved. (F) On the other hand, *rsc8*-depleted *chd1Δ* loses the regular wave-like shape of its second fPC after the +2 nucleosome to form broader peaks, indicating the presence of gene-specific nucleosome profiles as in WT conditions. All axes are scaled to the same size for each strain; shapes and amplitudes are therefore comparable (see Methods for more details).

hypothesised that the arrangement is strictly limited to the gene body. Indeed, the second small-gene fPC indicates well-defined positioning only for up to the +2 nucleosome (≈300 bp), and the function loses quickly its frequent wave-like shape thereafter (Fig 2D). The two major fPCs for small genes are not sufficient anymore to separate the *all-gene* clusters, which are discriminated by the presence of downstream nucleosomes. To verify our hypothesis of gene-size dependent phasing, we divided the regions into small and large genes *before* performing the Pearson clustering. When considering exclusively small genes, the two Pearson groups become linearly separable again, which is—in accordance with our hypothesis—predominantly determined by the size (S6 Fig). This shows that the nucleosome arrangement is strictly limited to the gene body.

The data produced by [12, 18] contain two replicates for *chd1Δ*, *isw1Δ*, and *isw2Δ* cells as well as *rsc8*-depleted strains, together with their combinations as double, triple, and quadruple

mutants. In order to analyse how gene-size dependent nucleosome phasing alters in varying contexts, we compared the small-gene fPCs in mutant and WT conditions. Surprisingly, the separation of the *all-gene* clusters was clearly visible for the fPCs of small coding regions in *rsc8*-depleted strains (Fig 2B). Indeed, the average MNase-seq profile exhibits phased peaks along the entire 1000 bp-window (Fig 2E), and nucleosome positioning continued outside the gene boundaries (S4 Fig). The linear separability of the *all-gene* clusters using small-gene fPCs can be found in almost all mutants which are depleted of Rsc8 (S7 Fig), with the sole exception of Rsc8-depleted *chd1Δ* strains (Fig 2C, replicate B S8(B) Fig). Here, the groups cannot be visually separated by $\zeta^1$ and $\zeta^2$, and the determined fPCs resemble small-gene fPCs in WT conditions (Fig 2F, replicate B S8(D) Fig). This indicates that the gene-specific boundaries for nucleosome phasing can be re-established, and the second fPC loses its wave-like shape again after the +2 position (S4 Fig). Consequently, we hypothesise that Chd1 and Rsc8 have partially antagonistic roles for maintaining chromatin organisation that distinguishes transcribed from non-transcribed regions. Taken together, this analysis exhibits strictly constrained and Rsc8-mediated nucleosome organisation within coding regions.

## Nucleosome phasing changes in remodeler mutants

We were particularly interested in how nucleosome remodeler complexes affect coordinated phasing. To remove the gene size-dependent bias from the clustering and the established fPCs, we applied the Pearson clustering to exclusively large genes (>1000 bp) for all strains and determined their two major fPCs (S2C, S2D and S5 Figs). We can confirm that the created groups for all mutants were again significant (i.e. average p-value of a KS test with 500 random partitions was lower than 5%, see S9 Fig). Interestingly, the Pearson clusters were always visually separated by using solely the first two fPCs, although some strains exhibited a larger overlap between the groups than others (S10 and S11 Figs for replicate A and B, respectively). This suggests that coordinated phasing in all mutants can be represented by considering only the two fPCs that describe the most variance, and including more fPCs is not necessary in order to interpret the discriminating function.

The respective contribution of the two major fPCs to separate the clusters varied between the strains, suggesting that fPCA is sufficiently sensitive to capture strain-dependent consequences (S10 and S11 Figs for replicate A and B, respectively). This caused the slope of the discriminating boundary to tilt. Therefore, the transformations of the mean distribution (i.e. fPCs and their factors $\zeta_i^j$) changed for these strains. This indicates that they had not only a global effect on the average MNase-seq profile, but also caused a gene-specific disruption of the nucleosome positioning. We deemed those strains particularly important that altered the gene-specific collective behaviour of the *entire* nucleosome array with respect to the WT. To clarify this point, we provide an example for the impact of a fictional mutant that only affects the mean but not the ensemble of nucleosomes as a whole in S12 Fig. We determined the slope for all strains using a linear SVM. As aforementioned, the boundary is tilted when only considering large genes in WT conditions (Fig 3A), and the two available replicates differ slightly. The observed deviation between replicates was used as a reference for the anticipated variability in the data. By using Eq 7, we determined chromatin remodeler-deficient strains that had a sufficiently different linear boundary with respect to the WT.

We provide three different perspectives on the data. Firstly, the cluster distribution with respect to the factors $\zeta_i^j$ together with the slope highlight mutants that particularly disrupt gene-specific collective nucleosome phasing. In the following, clusters are always indicated using the colours orange and blue. Secondly, analysing the transformation of the two major fPCs of the mean unlocks an additional understanding of the variance present in the data and

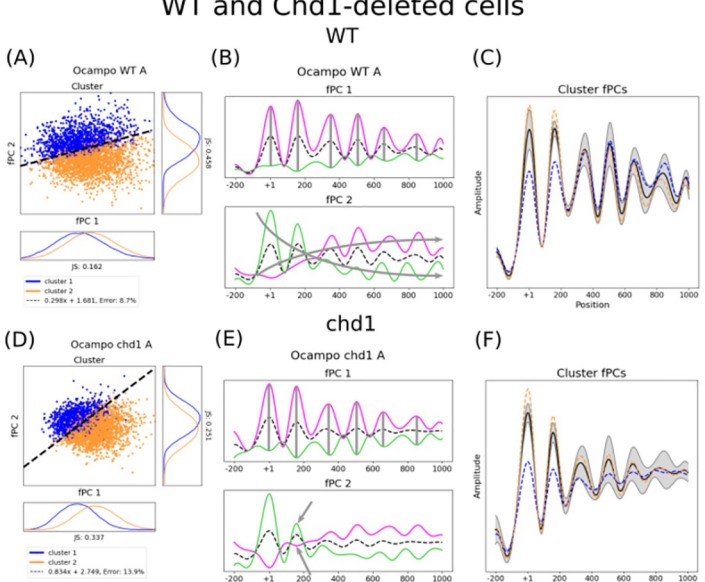

**Fig 3. The fPCs, their gene specific scores, and the discriminating boundary explain collective phasing and how this changes in *chd1Δ* with respect to WT conditions.** The figure shows the cluster distribution with respect to $\zeta_i^j$, the impact of the determined fPCs, and the location-specific impact of the separating boundary for WT (i.e. panels A-C) and *chd1Δ* conditions (i.e. panels D-F). Panels A and D show the fPC scores $\zeta_i^j$ of WT and *chd1Δ* strains, respectively. For the latter, the boundary slope changed notably (black dashed line). As indicated by the fCPs in panels B and E for WT and *chd1Δ*, respectively, the functional description of the data changes. Indeed, the second fPC of *chd1Δ* abates quickly after the +1, with a strong effect on the effect of the +2 (grey arrows). The dashed black line as well as the solid lines in magenta and green indicate the mean, a positive contribution of the fPC, and a negative contribution, respectively. When exact positions were seemingly not affected by the fPC, we marked the positions with a grey vertical bar. General trends are given in grey arrows along the gene. The location-specific impact of the separating boundary is given in panel C for WT and panel F for *chd1Δ* strains. Interestingly, despite the median distributions of the clusters (blue and orange) are clearly different with respect to the +1 and +2 in WT conditions, later positions are much more important for allocating a profile to a particular group (grey areas, mean in black). Whilst this is also true for *chd1Δ*, the importance of later nucleosomes is even more accentuated, whereas the influence of the +1 and +2 positions are further decreased. All axes are scaled to the same size for each strain; shapes and amplitudes are therefore comparable (see Methods for more details).

allow quantifying the general impact of chromatin remodeler deficiency. Here, we plot a positive contribution to the mean in magenta and a negative contribution in green. Lastly, the location-specific effect of the discriminator links spatial properties to the Pearson gene clusters, which describe likely similarity of nucleosome positioning mediated by chromatin remodelers. The impact of the discriminator is in the following given as a grey area around the mean, indicating more important regions when the margin is larger. The median profile of each cluster using the determined fCPs will be given again in orange and blue. This allows a comprehensive analysis of the impact of gene deletions or *rsc8* depletion with respect to the WT.

We can determine the importance of particular positions to separate the clusters as follows. The slope of the SVM indicates the contribution of each fPC to separate the clusters. For example, a 0° angle shows that the descriminator can be solely described by the second fPC; 45° suggest an equal contribution of both fPCs to separate the clusters; and 90° indicate that collective nucleosome phasing is exclusively dependent on the first fPC. Consequently, by linearly combining both fPCs together as implied by the slope (Eq 9), we can evaluate which positions along the profile are particularly important for the classification. Indeed, understanding the separating boundary is not straightforward. Although the median profiles for each profile can

**Table 1. SVM boundary slopes for both replicates.** The first two rows give the boundary slope for replicate A and B, respectively. Mean $\mu$ is the mean slope for both. The **s** value represents our significance measurement defined in Eq 7. Noteworthy changes of the boundary slope are marked in green (bold), all others are red. The s-value in WT is per definition equal 0.

|          | WT    | chd1   | isw1   | isw2   | rsc8   | isw1/chd1 | isw2/chd1 | chd1/rsc8 |
|----------|-------|--------|--------|--------|--------|-----------|-----------|-----------|
| **A**    | 0.299 | 0.834  | 0.117  | 0.133  | 0.377  | 0.213     | 1.406     | 0.031     |
| **B**    | 0.055 | 0.48   | 0.329  | 0.038  | 0.08   | 0.283     | 0.538     | 0.074     |
| **Mean $\mu$** | 0.177 | 0.657 | 0.223 | 0.0855 | 0.2285 | 0.248 | 0.972 | 0.0525 |
| **s**    | 0     | **2.6674** | 0.0409 | 0.3612 | 0.0366 | 0.2951 | **2.9842** | **1.4773** |
|          | **isw1/isw2** | **isw1/rsc8** | **isw2/rsc8** | **isw1/isw2 chd1** | **isw1/chd1 rsc8** | **isw2/chd1 rsc8** | **isw1/isw2 rsc8** | **isw1/isw2 chd1/rsc8** |
| **A**    | 1.452 | 0.347  | 0.153  | 0.112  | 0.216  | 0.057  | 0.466  | 0.066  |
| **B**    | 1.074 | 0.072  | 0.567  | 0.295  | 0.207  | 0.068  | 0.245  | 0.174  |
| **Mean $\mu$** | 1.263 | 0.2095 | 0.36 | 0.2035 | 0.2115 | 0.0625 | 0.3555 | 0.12 |
| **s**    | **12.7873** | 0.0157 | 0.3315 | 0.0157 | 0.5420 | **4.8846** | 0.5909 | 0.1233 |

differ substantially at some positions, this variance might be less important for separating and interpreting the clusters (e.g. the +2 nucleosome in WT conditions, Fig 3C). Reciprocally, whilst the median profiles for both groups can be very similar, the variance over all considered genes at this locus could be much larger and therefore play an important role for the classification (e.g. the +3 position in WT strains Fig 3C).

We identified 5 mutants—namely *chd1Δ*, *isw2Δchd1Δ*, *rsc8*-depleted *chd1Δ*, *isw1Δisw2Δ*, and *rsc8*-depleted *isw2Δchd1Δ*—that evoked notable changes considering the experimental variability between replicates (Figs 3–5). For a correct interpretation of the results, it is crucial to highlight that this does not imply that other mutants had no effect on the profile. Rather, this suggests that the considered mutation caused a gene-specific change of nucleosome phasing regulated by chromatin remodelers, which we assume is represented by the deviance of the stochastic process (i.e. the variance to describe the MNase-seq profiles). Other gene deletions can have other impacts that do not disrupt the gene-specific collective positioning. All measurements are given in Table 1.

Most single mutants had only a small or negligible effect on the collective nucleosome phasing along transcribed regions, with the exception of *chd1Δ* (Fig 3D–3F). Indeed, the boundary was notably tilted with respect to WT conditions (Fig 3D). This suggests that the functional composition of the MNase-seq signal changed. In fact, the amplitude of the second fPC decreases more quickly along the gene body in *chd1Δ* mutants, and the variance of the peak at the +2 position strongly diminished (Fig 3E). When interpreting the effect of the discriminating boundary, we observe that the +1 and +2 nucleosomes only exhibit a small importance for establishing the clusters, whereas the impact of the NDR and later nucleosomes increased (Fig 3F). Consequently, the +1 position remains largely unaffected. As Chd1 is responsible for nucleosome spacing along genes and is particularly involved in maintaining chromatin integrity during Pol II elongation, it is intuitive that the *chd1*-deletion influences phasing within the gene body. This outcome shows the clear effect of chromatin maintenance by Chd1 after the +2 nucleosome, whilst leaving the +1 position well preserved.

The double mutant *isw2Δchd1Δ* exhibited also a noteworthy shift of the separating boundary (Fig 4A), yet with different results to the *chd1Δ* single mutant. The second fPC seemingly preserves its wave-like shape (Fig 4B). This indicates that nucleosome presence is less perturbed, and peaks are comparatively well positioned. Similar to the *chd1Δ* single mutant, both of the fPCs strongly contribute to distinguish between the Pearson clusters. The discriminating function exhibits similar local effects as the *chd1Δ* strain, but the positions after the +2 nucleosome clearly indicate an additional shift which contributes to the separation (Fig 4C).

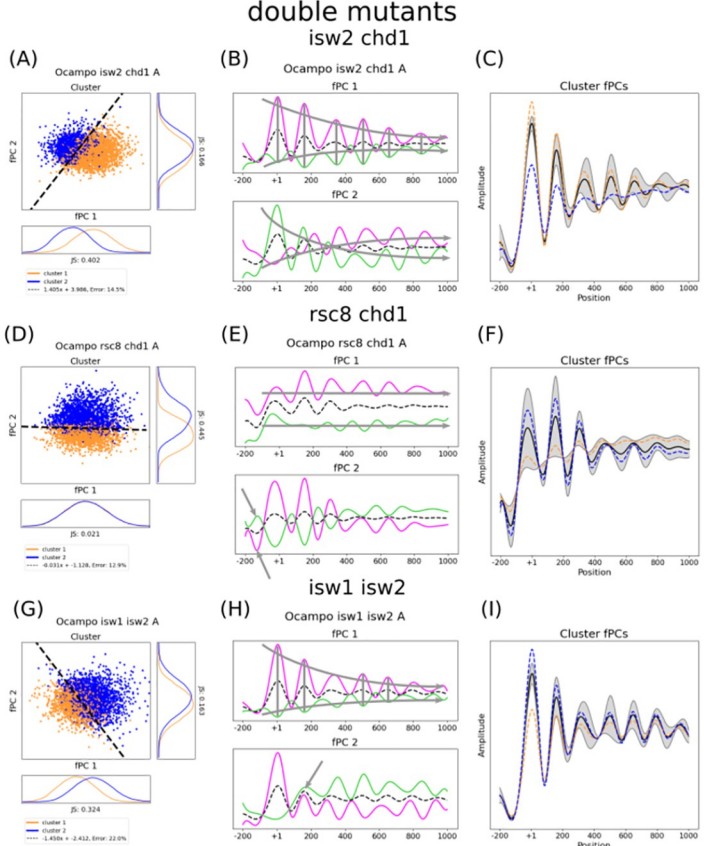

**Fig 4. The fPCs, their gene specific scores, and the discriminating boundary explain changing collective phasing in double mutants.** The figure shows the cluster distribution with respect to $\zeta_i^j$, the impact of the determined fPCs, and the location-specific impact of the separating boundary for all double mutants, in particular *isw2Δchd1Δ* (i.e. panels A-C), *rsc8chd1Δ* (i.e. panels D-F), and *isw1Δisw2Δ* (i.e. panels G-I). The linear separation of the cluster distribution with respect to factors $\zeta_i^j$ indicate a notable gene-specific change for the three mutants in panels A, D, and G. The two clusters are given in orange and blue, and the SVM boundary is depicted by the black dashed line. Whilst *isw2Δchd1Δ* and *isw1Δisw2Δ* require both fPCs to linearly separate the Pearson clusters, *rsc8chd1Δ* is almost exclusively dependent on the second fPC, which means this mutant decreased the slope tilt. This can be better understood when analysing the two fPCs and their effect on the mean ((B) for *isw2Δchd1Δ*, (E) for *rsc8chd1Δ*, and (H) for *isw1Δisw1Δ*). The solid lines in magenta and green in these plots indicate a positive contribution of the fPC and a negative contribution, respectively, whereas the black dashed line depicts the mean. Grey arrows along the gene suggest general trends. Grey vertical bars suggest positions that remain largely unperturbed by the fPC. Grey arrows pointing to a single peak suggest remarkable properties. Interestingly, whilst the first fPC of the *isw2Δchd1Δ* and *isw1Δisw2Δ* strains shows a similar transformation of the mean, the second fPC indicates a different behaviour, particularly with respect to the +2 nucleosome. As suggested by the fact that clusters in the *rsc8chd1Δ* mutant are exclusively dependent on the second fPC, the first fPC explains only the average profile amplitude and does not contain any information about collective phasing. The location-specific effect of the linear separator for each mutant is given in (C), (F), and (I). The grey areas indicate the importance of each position to determine the clusters, whose median profile is shown as a blue and orange dashed line. The mean is depicted in black. Although the impact on the grouping of the +1 and +2 position in *isw2Δchd1Δ* conditions is similar to the *isw1Δisw2Δ* strain, the latter is seemingly particularly dependent on the +3 and +4 nucleosome. Positions thereafter become less important, which keep having a strong impact on clustering in *isw2Δchd1Δ*. As expected *rsc8chd1Δ* is exclusively dependent on the second fPC. Interestingly, the entire profile seems to be influential for classifying genes, with the largest impact allocated to the first two nucleosomes. All axes are scaled to the same size for each strain; shapes and amplitudes are therefore comparable (see Methods for more details).

Interestingly, *rsc8*-depleted *chd1Δ* significantly decreases the slope tilting (instead of accentuating it), therefore making coordinated phasing almost exclusively dependent on the second fPC (Fig 4D). This can be better understood when analysing their respective effects (Fig 4E). The first fPC solely explains average signal amplitude (which is not measured by the Pearson

correlation index) and hence contains almost no information about coordinated positioning. As expected, the local effect of the discriminating boundary follows the trend described by fPC2 (Fig 4F). The second fPC also indicates that the NDR before the +1 cannot be maintained (see arrow in Fig 4E and the grey area in NDR and +1 position in Fig 4F), which is in line with other studies [12, 40]. Remarkably, all nucleosome positions along the entire array seem to be important for the classification—particularly the first two—which is not the case for the other two double mutants. It should be noted that not all double mutants that include *chd1Δ* show a similarly notable tilting of the slope as the single mutant. This could possibly mean that these double mutants have opposing effects, although it is difficult to give a clear indication with the variation between only two replicates. We found an interesting behaviour for *isw1Δisw2Δ* (Fig 4G). The effect of the second fPC hints that the positioning of the +2 is strongly impacted, and following phasing becomes inharmonious (Fig 4E). The +1 is kept well positioned. The first fPC, on the other hand, resembles the first fPC of the *isw2Δchd1Δ* mutant, with a minor difference at the +3 nucleosome (compare Fig 4E with Fig 4B). Indeed, when analysing the location-specific properties of the separating function (Fig 4I), nucleosome profiles in the *isw1Δisw2Δ* strain seem to be clustered particularly with respect to a shift at the +3 and +4 position. This shift is apparently slightly corrected thereafter and becomes less important. Whilst seemingly similar, a shift in the *isw2Δchd1Δ* strain after the +2 position remains important for the entire arrangement to determine the gene groups (compare Fig 4I with Fig 4C). This indicates that Chd1 and Isw1 contribute differently to nucleosome phasing in *isw2Δ* conditions, with the effect of Isw1 being possibly more confined. Taken together, these results show that double mutants can have varying and non-linear effects.

Among the triple and quadruple mutants, the only one that changed notably the clustering boundary is *isw2Δchd1Δrsc8* (Fig 5A). Once again, tilting is decreased. The effect of the fPCs and the separating boundary is almost identical to the *chd1Δrsc8* mutant, suggesting that *isw2Δ* does not have a strong effect on the phenomenon (Fig 5B and 5C). However, it should be mentioned that the variability between the two replicates is considerably large, as the two clusters can be only neatly separated in replicate *B*, whereas replicate *A* exhibits a great overlap. Whilst the result in the latter replicate could suggest that more fPCs are necessary to interpret

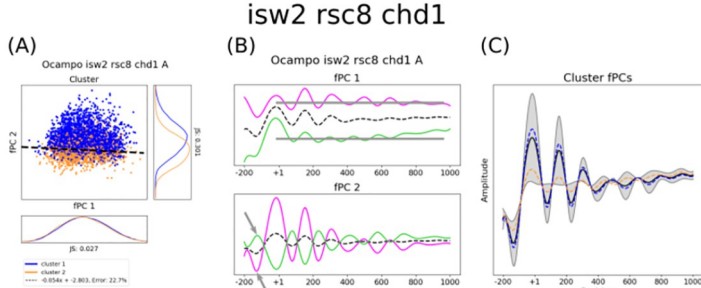

**Fig 5. The fPCs, their gene specific scores, and the discriminating boundary explain changing collective phasing in *isw2Δrsc8chd1Δ*.** The two clusters are given in orange and blue. The figure shows the fPC scores ζ of the *isw2Δchd1Δrsc8* mutant and their separating boundary (black dashed line, A). The slope decreases with respect to the WT, making the gene groups almost solely dependent on the second fPC. Both fPCs transform the mean in a similar way as the double mutant *rsc8chd1Δ* (compare panel B with Fig 4E). The dashed black line as well as the solid lines in magenta and green indicate the mean, a positive contribution of the fPC, and a negative contribution, respectively. As expected, the separating boundary discriminate between the two clusters largely following the second fPC (C). The grey areas show the importance of each position to determine the clusters, whose median profile is shown as a blue and orange dashed line. The mean is depicted in black. All axes are scaled to the same size for each strain; shapes and amplitudes are therefore comparable.

the gene groups, the results for replicate *B* indicate that sufficient information is preserved in the first two fPCs. More replicates would be needed to provide an answer. We also want to highlight that mutants with more than two gene deletions exhibited less clear nucleosome peaks, and a straightforward interpretation of the Pearson correlation with respect to the two discriminating trends (compare with cartoon in Fig 1D) could be difficult. The results for these strains should be taken with a pinch of salt.

Taken together, these outcomes show that remodeler mutants have varying effects on nucleosome positioning. Whilst most mutations do not notably alter the gene-specific nucleosome coordination with respect to the WT, we identified 5 mutants that exhibited a strong effect on phasing. Interestingly, most of them include *chd1Δ*, which indicates an important role of Chd1 for local arrangement within the gene body. Using fPCA to visualise the Pearson clusters permits the clear and position-specific quantification of the induced impact among varying strains.

## Pol II presence correlates with nucleosome organisation in *chd1Δ* mutants

In order to assess an interdependence of nucleosome organisation with other genomic properties, we compared the two Pearson clusters to Pol II levels, Sth1 occupancy, AT ratio over the entire gene, as well as upstream NDR length and orientation of the upstream NDR (i.e. tandem or divergent). We also included Mediator presence as a large protein complex with transient interactions predominantly at the NDR. All of these factors were clustered into two equally-sized groups where possible. For example, Pol II presence along the gene was evenly separated into transcribed regions with high and low Pol II occupancy. Interdependence was measured by training a simple neural network with no hidden neurons using Hebbian learning [41]. Consequently, we assessed which nuclear groups (e.g. high or low Pol II presence) corresponded to which Pearson clusters. We want to stress that we did not aim to find a predictive model. Rather, this approach allowed us to measure a direct correlation between similarity of nucleosome phasing and other genomics properties. The initial *k*-mean clustering did not impose a constraint on the group size, and they could therefore differ in the number of genes they contained. To remove any prediction bias, we forced the clusters to be of the same size. Genes in the larger group with closest Pearson coefficient to all distributions in the smaller group changed the cluster. We found the analysis for WT conditions non-conclusive, and correlations varied between *A* and *B* replicate (S13 Fig top). Whilst *A* was slightly correlated with the AT sequence content (Fig 6A and 6B), this trend disappeared for *B*, and it might in both replicates rather correspond to the fPC orthogonal to the cluster boundary (S16 Fig). Overall, we were surprised that none of the investigated properties could indicate a clear interdependence with nucleosome phasing in WT (Fig 6A).

The correlation between positioning and other nuclear properties changed among the mutants (S13 Fig). The effect is particularly clear for *chd1Δ* (Fig 6C), as there is a strong interdependence between phasing and Pol II (Fig 6D), Mediator presence, and NDR size (S13 Fig). As aforementioned, Chd1 maintains, among others, chromatin integrity during Pol II elongation. The correlation is therefore in line with our previous conclusions and the function of Chd1. The established link between Pol II presence and nucleosome organisation remains conserved in all strains with a Chd1 gene deletion, except *isw1Δchd1Δrsc8*. This is similarly true for the correlation with Mediator occupancy and NDR length. There was also a slight correlation to Sth1 and AT ratio in cells containing *chd1Δ*, which were, despite being weak, still notably stronger than in WT. The results are in agreement with the effects of Chd1 on chromatin maintenance during gene expression.

Due to the Rsc8-mediated nucleosome organisation within transcribed regions, we were wondering whether there is an increased interdependence to NDR length or gene size. We can

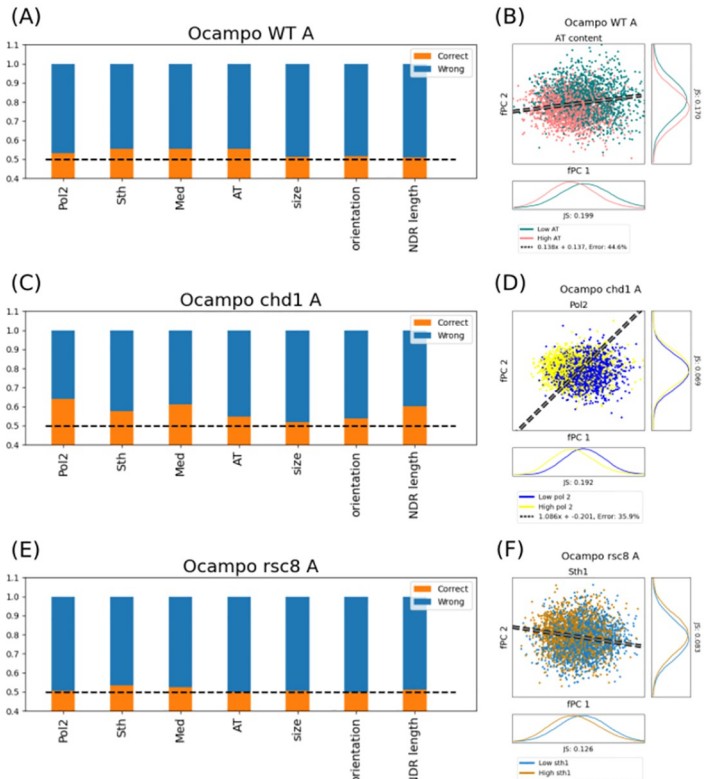

**Fig 6. Remodeler deletions have varying effects on the interdependence with other genomic properties.** The orange bars in panels A, C, and E show the ratio of correct predictions, and blue bars are wrong guesses. As we distinguish between two clusters, the dashed black line at 0.5 indicates random guessing. The dashed grey line with black edging in panels B, D, and F display the linear boundary for the Pearson clusters. Panels A and B: WT conditions are seemingly correlated with the sequence composition. However, the results are different for the *B* replicate, and therefore non-conclusive. All possible correlations are surprisingly low. Panels C and D: *chd1Δ* mutants increase particularly their dependence on Pol II and other transcription-coupled properties, such as Mediator presence. Surprisingly, the mutant showed also an increased interdependence on NDR length. Panels E and F: despite the Rsc8-mediated gene limits, there is no correlation with coordinated nucleosome phasing and the size of transcribed regions or NDR length. Although Sth1 indicates a slightly increased interdependence, this cannot be confirmed when plotting the groups with respect to the Pearson clusters. This is in line with our hypothesis that positioning in different regions interfere, and therefore, nucleosome localisation become increasingly independent from region-specific factors.

report that there is no correlation with NDR size in any *rsc8*-depleted strain (Fig 6E). This is in line with our hypothesis that Rsc8 decouples processes at different genes. However, *rsc8* mutant cells exhibited a slightly increased correlation with Sth1. By looking at the separation with respect to the Pearson cluster boundary, we find that there is no noticeable impact (Fig 6F). The results indicate that any correlation with region-specific properties is lost, which is likely due to the interference of nucleosome positioning of various regions.

*isw1Δ* single mutant did exhibit only a slightly increased correlation with Pol II, Sth1 and Mediator presence as well as AT ratio. *isw2Δ* might as well show a weak correlation with Pol II occupation. Each of the replicates of their double mutant indicates different correlations, and it is therefore difficult to tell whether transcription-related factors influence nucleosome phasing in the *isw1Δisw2Δ* strain. However, none of them indicate any strong interaction, suggesting that—on a global scale—these effects might be negligible in comparison to the WT (S13 Fig).

Interestingly, the *rsc8*-depleted *isw2Δ* indicated a strong correlation with Sth1 and Mediator presence as well as NDR length. The effect was observable in almost all strains that contained

the double mutation with the exception of the quadruple mutant (S13 Fig). Taken together, this could indicate an impact along the gene body and the promoter region in strains that contain *rsc8*-depleted *isw2Δ*.

Surprisingly, combining two factors together (e.g. Pol II presence and AT ratio) to predict Pearson clustering did not increase accuracy. Instead, one factor dominated the correlation measurement, e.g. Pol II presence for *chd1Δ* strains. This could possibly suggest that—despite several factors showing increased interdependence—they can be reduced to a main influencing factor (which is not necessarily one of the tested properties).

Taken together, the results indicate a strong interdependence between local genomic properties—such as presence of large protein complexes or NDR length—and strains containing *chd1Δ*. This supports our hypothesis of Chd1 being responsible for local nucleosome coordination.

## Discussion

In this work, we analysed the collective positioning of nucleosome arrangement within the gene body in WT and chromatin remodeler-deficient strains by combining clustering of Pearson coefficients with fPCA, the latter being an analysis framework for functional data. Although fPCA is well established in the assessment of time series, it has not been previously used to understand location-specific nucleosome profiles on a global scale. As we argue that the Pearson index measures similarity of nucleosome arrangement between genes, we interpreted the effect of chromatin remodelers on the positioning by visualising the distribution of two established significant Pearson clusters using fPCA. Indeed, we can show that the sets of genes for all mutants can be sensibly separated by the two fPCs that explain most variance in the data, and more fPCs are not necessary to describe the clusters. This allowed the quantification of the effect on coordinated phasing. The significant Pearson groups were compared with other nuclear properties—such as Pol II presence and NDR maintenance—and sequence-dependent characteristics. None of the commonly supposed influencing factors can easily explain coordinated nucleosome positioning in WT conditions. However, correlation between tested properties and phasing increases with some gene mutations. The analysis reveals the impact of different gene deletions of chromatin remodelers on nucleosome arrangement within the gene body. It shows Rsc8-defined boundaries for nucleosome positioning along the gene, suggesting a global impact over the entire array for each gene. On the other hand, the results for most strains that contained a Chd1 deletion indicated gene-specific local effects, which correlate largely with Pol II presence. In the following, we critically discuss the results and their significance.

We applied a pairwise Pearson cross-correlation index to measure profile similarity between genes. The linear correlation measurements evaluate the overall trend of the signal (i.e. increasing or decreasing distributions at similar positions), and it does not take signal scaling into account. Therefore, it assesses whether genes are apt to contain nucleosomes at similar positions. Indeed, similar nucleosome phasing could indicate similar but gene-specific chromatin remodeler dynamics, which justifies the rationale for measuring classical linear correlation. It also follows previous analyses using comparable measurements [33–35].

We classified genes according to their Pearson similarity by applying a $k$-mean clustering approach. $k$-mean was repeated over several random initialisations, therefore removing any prior bias. We used a silhouette criterion value to determine the best number of clusters, which was shown to be 2. It should be mentioned though that the cluster distribution according to the fPCA did not show a clear separation of the data points themselves (i.e. there were no distinct data accumulations). Thus, this clustering is imposed by our assumptions using the Pearson index. Nonetheless, we argue that they reveal important information about

nucleosome phasing linked to chromatin remodelers when compared with mutant strains. Grouping of nucleosome profiles has been previously performed using different algorithms [42], and it is necessary to show that a separation along the first two fPCs is similarly possible when using other clustering approaches. We therefore compared the results of the Pearson $k$-mean clusters with a WARD algorithm using the Euclidean distance metric. Once again, profiles can be best separated into two clusters (S14(A) Fig). Moreover, the gene groups tend to separate along the first fPCs, despite the fact that the boundary is less neat (S14(B) Fig). It should be mentioned though that clustering with an Euclidean distance measures different properties of the profiles, and it is therefore expected that the separation border changes. Indeed, the separation occurs now predominantly with respect to fPC1 (S14(B) Fig), which indicates a strong influence of the signal scaling that was ignored by the Pearson correlation. Therefore, the clustering does not express similarity of the overall nucleosome positioning along the entire array. We want to emphasise that we were particularly interested in how the entire nucleosome array tends to behave *as a whole*, which should be sensibly measured by using the Pearson correlation. We decided to ignore the scaling of the sequencing amplitude, as it conveys how many cells contain a single well-positioned nucleosome instead of how nucleosomes tend to behave with respect to most cells. Nonetheless, the found gene sets based on a fundamentally different clustering method tend to separate into different areas when using the first two fPCs. We conclude that the observed separation using $k$-mean clustering and the Pearson correlation as a distance metric captures genuine biological properties.

The determined gene groups were as different as possible using similarity measured by Pearson correlation. The validation using the silhouette criterion together with the KS significance test over the inter-cluster correlation between the found $k$-mean clusters and random grouping showed their significance. This shows that the data could not be better categorised using linear correlation. Significance tests are often prone to attributing a high importance to small differences when sample sizes are large. Therefore, we randomly sub-sampled 500 inter-cluster correlation indices for the determined and random grouping, respectively. It should be mentioned that in a few single instances, the p-value was larger than 5%, as we randomly selected 500 similarly low values for both inter-cluster correlations (i.e. for the $k$-mean and the random clusters). We accounted for these fluctuations by averaging the p-value over 500 random clusters from which each 500 random sub-samples were selected. Using this approach, we found that the $k$-mean clusters were significant for large genes of all mutants (average p-value <5%).

As the Pearson correlation index only indicates average similarity over the entire nucleosome array, we aimed to compare the clusters to the data itself in order to interpret their differences. Dimensionality reductions are often used to visualise clusters, such as for single-cell sequencing analyses [43]. Common approaches include PCA, uniform manifold approximation and projection (UMAP) [44], and t-distributed stochastic neighborhood embedding (t-SNE) [45]. Whilst the latter two are non-linear dimensionality reductions, PCA and fPCA find a linear decomposition of the data into the axes (or functions) that explain most variance. It is challenging to retrieve the exact meaning of the discriminating boundary using non-linear approaches. Consequently, understanding the location-dependent differences in the profile between two clusters and interpreting their separating function is more straightforward for PCA or fPCA than for UMAP and t-SNE. Although PCA and fPCA are very similar, PCA assumes that every position in the MNase-seq data is independent, whereas fPCA conjectures that they were produced by a single stochastic process along the spatial axis. Therefore, positions are dependent on each other. This is in line with the barrier model for establishing nucleosome phasing, which makes fPCA preferrable over PCA. Moreover, as we treat each nucleosome profile as one entity by using the Pearson correlation, the independence

assumption would violate the fundamental understanding in our analysis. Nonetheless, when comparing PCA and fPCA, we showed that the two clusters can be similarly separated (S15 Fig), although the two principal axes are slightly differently shaped due to the missing constraint of the spatial dependence.

FPCA assumes a stochastic process with a mean behaviour over the entire data set, and it characterises each data point with respect to their deviance from that mean (see Methods). The results therefore depend on the entire considered data set. Indeed, we find different results when including all genes or exclusively transcribed regions >1000 bp. However, these differences are not strong. Moreover, any possible bias was excluded by removing genes smaller than 1000 bp from a subsequent analysis. Due to the abundant and well-positioned nature of nucleosomes within the gene body in *Saccharomyces cerevisiae*, we find it justified to presume an average nucleosome distribution describing their wave-like profile. Nonetheless, we argue that the variance between genes contains important information about nucleosome phasing imposed by chromatin remodelers, which we roughly categorised into groups. We found that the two Pearson correlation clusters could be neatly separated by the fPC scores $\zeta_i^j, j \in \{1, 2\}$. This indicates firstly that the Pearson index measures a trend that is explained by the largest variance in the data; and secondly, the two fPCs that describe most variance are sufficient to interpret the clusters.

Whilst linear-correlation measurements are limited to quantifying the average similarity, a combination with fPCA allows characterising location-specific differences and in which way gene deletions affect phasing from an average. Evaluating the effect of the linear boundary along positions within the gene body revealed detailed differences in the nucleosome profile that are important for establishing the groups. As our approach is largely dependent on general signal processing methods, we can repeatedly apply the same framework for all mutants and compare there results. Therefore, the combination of linear correlation with fPCA extends previous ways of analysing nucleosome distributions using only correlation measurements that average over the entire profile [33–35].

The analysis can clearly distinguish between mutant-specific effects on phasing. All mutants preserved the information of coordinated nucleosome arrangement in their first two fPCs, and the Pearson clusters could be separated by a neat line. Consequently, none of the chromatin remodeler gene deletions caused random positioning. Some mutants, however, showed an increased overlap between the two groups, which indicates increased independence between individual nucleosome locations, and positioning might be more random. Including more fPCs could help further separating the clusters. In all of those cases however, one of the two replicates always permitted a clearer separation by using only the first two fPCs. Considering the experimental variability in the data, it is not possible to draw direct conclusions without further replicates. In order to simplify the comparison between mutants, we restrained from including more fPCs.

Most strains did not alter notably their gene-specific collective arrangement (i.e. the slope), and a linear separation of the Pearson clusters using the deviance from the mean did not change with respect to WT strains. Although they can nevertheless have an impact on the mean itself, coordination along the genes remains preserved in a similar way, at least as measured by the Pearson correlation index. Due to the focus of the study on coordinated nucleosome positioning along transcribed regions, we did not consider them as having notably changed their coordinated phasing.

The length of an average yeast gene is approximately 1500 bp, which corresponds to 6–7 nucleosomes. To analyse the effect of nucleosome remodeler deletions or their depletion on their arrangement, we examined a stretch of 1200 bp per gene, including 200 bp upstream of

the +1 dyad. By considering an array of 6–7 nucleosomes at the same time, we were able to assess their long-range impact on the gene body. During our analysis, we differentiated between small and large genes (i.e. coding regions shorter than the considered 1000 bp after the +1 dyad versus genes $\geq$ 1000 bp). This revealed the effect of Rsc8 on limiting nucleosome positioning to the gene body. It should be emphasised that we removed smaller genes from the subsequent analysis, as it would be impossible to compare nucleosome arrangement further down into the gene body. We also surmised that the influence of remodeler deletions on later nucleosomes is more difficult to observe in smaller genes, since they could be located very right at the TTS which could have an additional impact. Therefore, we based our analysis and conclusions—particularly with respect to Chd1—only on genes larger than 1000 bp. It should be noted that the results may not be easily transferable to small genes.

Gene mutations of chromatin remodelers have been analysed previously in detail, including their influence on phasing [12–14, 18], NDR maintenance [46], and gene transcription [13]. RSC is the only essential chromatin remodeler complex in *Saccharomyces cerevisiae* [47], and it has been particularly associated with positioning of the +1 and -1 nucleosomes [12, 46, 48]. This mechanism has been proposed to be conserved among various yeast species [11]. It has also been reported that RSC regulates expression of Pol II and Pol III-transcribed genes [13, 49, 50]. Moreover, it has been found to impact Pol II elongation and termination [12]. All of these results imply that RSC is to some extent involved in limiting the transcribed region. However, this has been predominantly quantified with respect to changes at the core pro-moter. To our knowledge, a potential role for Rsc8 to decouple nucleosome phasing in inde-pendent genes has not been suggested. The presented functional analysis of MNase-seq profiles in *rsc8*-depleted strains clearly indicates a coordinated nucleosome arrangement that exceeds the limits of transcribed areas. This is further supported by our finding that correlation with other nuclear and sequence-dependent factors decreases. Furthermore, mutants that were *rsc8* depleted decreased notably the boundary slope between the two clusters, indicating that coordinated positioning becomes increasingly independent of other functional compo-nents. The strictly limited and Rsc8-mediated phasing barrier could have further implications for other processes—such as transcription—as nucleosome placing in one gene influences its neighbouring regions. The notion of gene-interfering positioning has been also proposed by [14]. The study shows that RSC could act as a bidirectional barrier, influencing upstream and downstream regions. Interestingly, they found that interference also plays a crucial role in WT strains, and that the same phenomenon remains preserved in *rsc8*-depleted cells. However, our fPCA reveals that the limiting role of the RSC remodeler complex is crucial in WT condi-tions, and that this behaviour is significantly altered when Rsc8 is depleted. Taking this into account, Rsc8 should fulfill the role of disentangling gene-related processes in WT strains, and it therefore allows for a flexible and uncorrelated transcriptional program. Indeed, *rsc8*-depleted cells exhibit significantly altered Pol II profiles [10, 12], which is in accordance with our hypothesis. We propose that the RSC chromatin remodeler globally disentangles nucleo-some phasing, and it therefore plays a substantial role in long-range positioning.

Interestingly, our results indicate that positioning limited to the gene body can be re-estab-lished in *rsc8*-depleted *chd1Δ* mutants. We hypothesise that they have antagonistic effects in establishing gene size-dependent barriers for nucleosome arrangement. Indeed, it was reported that Rsc8 and Chd1 have opposing effects for Pol II termination. *rsc8*-depleted cells exhibit inhibition of Pol II dissociation at the TTS, whereas the double mutant *isw1Δchd1Δ* increases release frequency, with seemingly *chd1Δ* dominating this effect [12]. The authors propose that this is related to the close packaging of nucleosomes at the TTS. Our outcomes suggest that they might have antagonistic effects in chromatin organisation that differs between transcribed and non-transcribed regions.

We found that *chd1Δ* mutants had a strong impact on coordinated positioning within the gene body. Indeed, Chd1 has been, among others, characterised with respect to its role in maintaining chromatin integrity during Pol II transcription [16, 51, 52], and it associates to both promoters and transcribed regions [53]. This is in line with our finding that correlation with Pol II presence and occupancy of Mediator increases in Chd1-deficient strains. With the exception of *isw1Δisw2Δ*, all other noteworthy changes included deletion of *chd*, further emphasising its role for chromatin organisation within the gene. However, not all *chd1Δ*-containing mutants exhibit a notable effect. This can have various reasons, including experimental variability. However, particularly the mutant *chd1Δisw1Δisw2Δ* could indicate an interacting behaviour of the remodelers. Indeed, Chd1 has been reported to cooperate [16] as well as antagonise Isw1 [18], and therefore could have different effects depending on the context. With this being said, the behaviour of the triple mutant *isw1Δisw2Δchd1Δ* is particularly interesting, as *chd1Δ* and *isw1Δisw2Δ* each individually affect coordinated phasing, but not their triple mutant. This could suggest an antagonistic behaviour on nucleosome coordination. As Chd1 is highly conserved in all eukaryotes [54], this result could have consequences beyond *Saccharomyces cerevisiae*.

It is worth noting that the Isw1 subunit is part of the two complexes Isw1a and Isw1b, and consequently its deletion affects them both. Previous studies [55] have shown that these complexes have different functions and enrichment profiles. The Ioc3 subunit of the Isw1a complex is enriched at +1 nucleosome, while the (Isw1b) is enriched on +2, +3, and +4 nucleosomes. In addition, these two complexes can act on different sets of genes and promote sliding in different directions. However, as we did not analyse the effect of specific subunits of Isw1a or Isw1b complexes in this study, we cannot distinguish between the two complexes. Therefore, it is possible that the effects observed from the +2 nucleosome onward may not be due to one specific complex.

Analysing the MNase-seq data using fPCA allowed us to obtain a different view on the functionality of various remodelers to maintain chromatin organisation. We propose the following mechanism (Fig 7). The RSC remodeler complex is essential for allowing independent

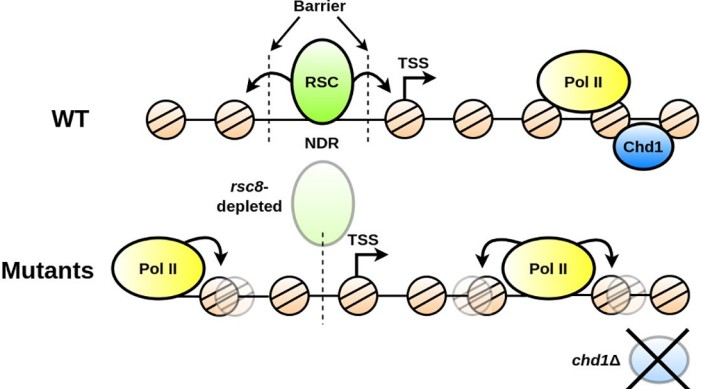

**Fig 7. Chromatin remodelers maintain nucleosome organisation on a local and far-reaching scale.** Top: RSC (green ellipse) establishes independent nucleosome phasing on each gene (two vertical dashed lines) by maintaining the NDR through positioning the +1 (cornered arrow) and -1 nucleosome. The ATP-dependent positioning is symbolised by black arrows pointing away from RSC. The local remodeling effect of Chd1 (blue ellipse) allows chromatin arrangement independent of Pol II transcription (yellow ellipse). Bottom: in *rsc8* strains, the NDR cannot be maintained anymore, and phasing in and outside a gene interfere with each other (single dashed line). We propose that this should equally lead to an increased interdependence of other nuclear processes such as transcription. If *chd1* is deleted, nucleosome arrangement is more sensitive to the presence of other large complexes, such as Pol II. During transcription, Pol II is affecting the local positioning (black arrows from Pol II).

phasing in each single gene. It plays therefore a pivotal role in maintaining the barrier with respect to which nucleosome positioning is coordinated. This permits the decoupling of gene-specific processes such as transcription. Depletion of Rsc8 leads to the interference of different genomic regions, which therefore alters sequence accessibility on a global scale. Indeed, it has been reported that gene expression is dramatically changed in *rsc8* mutants [10, 50]. Chd1, on the other hand, maintains chromatin integrity during transcription [16, 51, 52], and it influences nucleosome phasing locally to permit Pol II-mediated expression. *chd1Δ* strains make positioning dependent on Pol II presence. Consequently, whilst RSC plays a global role, Chd1 is important for local nucleosome organisation.

## Methods

### Data treatment

MNase sequencing reads were taken from [18] and [12] (GEO accession numbers GSE69400 and GSE73428, respectively) and treated as in our previous study [56]. To be precise, reads from Fastq files were trimmed with `trim_galore` (v0.6.5) [57] and `cutadapt` (v3.1) [58]. Subsequently, they were mapped on the *Saccharomyces cerevisiae* genome (University of California at Santa Cruz [UCSC] version `sacCer3`) using `bowtie2` (v2.3.4.3) [59]. Files were converted with `samtools` (v1.9) [60] and `deeptools` (v3.5.0) [61]. Read counts were normalised in Reads Per Million (RPM) of mapped reads. We used the option `--MNase` of `bamCoverage` so that only the mononucleosome fragments were kept. This means that fragments shorter than 130 bp and longer than 200 bp were removed from analysis. Mediator and Sth1 ChIP-seq were taken from [56] (ArrayExpress accession number E-MTAB-12198). We used Pol II ChIP-seq from our previous study [62].

Following [12, 18], we retrieved positioning profiles along the coding regions 200 bp before and 1000 bp after the +1 nucleosome. Genes on the Crick strand were inverted. Consequently, all data is calibrated such that the +1 position is at 200 bp. The profile of genes for which the +1 position is known were considered as in [18].

### Measuring profile correlation and clustering

The pairwise Pearson correlation of MNase-seq distributions for each gene was determined using equation

$$r_{xy} = \frac{\sum_{i=1}^{n}(x_i - \bar{x})(y_i - \bar{y})}{\sqrt{\sum_{i=1}^{n}(x_i - \bar{x})^2}\sqrt{\sum_{i=1}^{n}(y_i - \bar{y})^2}}. \tag{1}$$

Here, $x$ and $y$ denote two genes, $\bar{x}$ and $\bar{y}$ symbolise their respective average MNase-seq value along the coding region, and $n = 1200$ is the length of the considered region. Eq 1 ranges between -1 and 1, and indicates whether the two gene profiles tend to change into the same (positive Pearson correlation) or opposite directions (negative Pearson indices).

Genes were divided using the *k*-mean clustering implementation in MATLAB with the correlation distance metric. To define the optimal number of *k*-mean clusters, we used the silhouette criterion measurement [63, 64]. For all analysed strains, the highest silhouette value occurs at 2 groups, suggesting that in order to divide the profiles into classes with respect to their Pearson indices, the optimal number of clusters is 2 (Fig 1A). Therefore data were grouped in two clusters (Fig 1B and 1C).

Cluster significance was validated by comparing the inter-cluster correlation (i.e. each pairwise Pearson index between a profile in cluster 1 and a profile in cluster 2) between the

determined and random clusters. It is expected to be minimal for the gene groups found by the $k$-mean clustering. Therefore, we applied the KS test implemented in `scipy` with the hypothesis that the inter-cluster correlation of the found gene sets is significantly lower than for random groups. It should be noted the number of all pairwise comparisons can be quite large. It is a known problem that small differences are marked as being significant by the KS significance test when sample sizes are too large. Therefore, we sub-sampled randomly 500 pairwise inter-cluster correlations for the determined and the random clusters, respectively. It should be mentioned that this sometimes led to very similar sub-samples, and the p-value was for some few runs fairly high. We therefore averaged the p-value over for the KS test over all 500 random clusters and used the conventional p-vale threshold of 0.05 (i.e. 5%) to reject the null hypothesis. To be more precise, we required that the average probability of finding random clusters that have a similarly low inter-cluster correlation was lower than 5%. This was the case for all tested mutants.

## Functional principal component analysis

Functional clustering in a Hilbert space $H$ can be achieved by fPCA. It applies—similar to PCA in Euclidean space—a functional dimensionality reduction in $H$ to investigate the dominant mode in functional data. Instead of relying on values in discrete dimensions, fPCA uses a given number of basis functions (e.g. B-splines or Wavelet) to create the eigenfunction basis that accounts for most functional variation. Despite the fact that MNase-seq data is stored in a discrete array (i.e. one value per bp), we can nevertheless find a functional approximation over a range using a given choice of basis functions. It should be noted that this implicitly smooths out high frequencies in the signal. We presume that nearby values in MNase-seq data possess a strong interdependence, therefore justifying a smoothed and continuous functional representation of the high dimensional data. In this study, we apply B-splines as a basis to represent the nucleosome array (Fig 8). We use the Python library `scikit-fda` to determine the fPCs and the corresponding weights explaining the distribution [65]. Here, we describe briefly the underlying principles of the method.

FPCA presumes that the functional data represents a stochastic process $X(t)$ with expected value $\mu(t) = \mathrm{E}[X(t)]$ and orthonormal eigenfunctions $\phi^i(t)$, $i = 1, 2. \ldots$. Intuitively, $\phi^i(t)$ describes the most variation in $X$ orthogonal to all $\phi^j$, $j < i$. This allows the iterative determination of the eigenfunctions in the functional data. It should be emphasised that in this study the process is defined in space rather than describing temporal data. We follow nevertheless the convention by denoting the independent variable as $t$. By using the Kosambi–Karhunen–Loève theorem, any stochastic process can be represented as an infinite linear combination

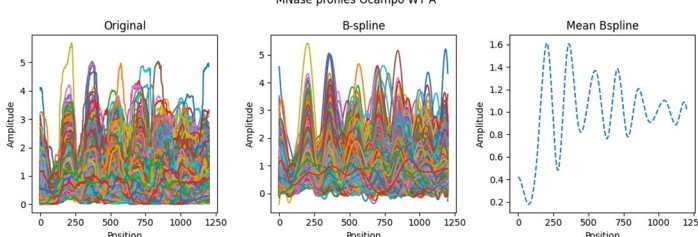

**Fig 8. Representing the MNase data array as a composition of B-spline base functions in WT conditions.** Left shows the raw data, each colour depicting one profile over a gene. Center gives the smoothed profiles after representing the data as B-splines. Right displays the average profile using the functional composition.

$\phi^i(t)$. Consequently, we can describe the stochasticity in $X(t)$ via

$$X(t) - \mu(t) = \sum_k \zeta^k \phi^k(t). \tag{2}$$

$\zeta^k$ is the autocovariance operator

$$\zeta^k = \int (X(t) - \mu(t))\phi^k(t)dt. \tag{3}$$

To provide some intuition, it is presumed that the entire data set can be explained via an average behaviour $\mu(t)$. Variability to $\mu(t)$ for each gene is expressed by $\phi^k(t)$ together with a factor $\zeta^k$. $\zeta^k$ can be loosely compared to a normal correlation measurement, i.e. $\zeta^k$ increases when $\phi^k(t)$ and $\int(X(t) - \mu(t))$ follow the same trend. If they describe opposing behaviours—for example $\phi^k(t)$ decreases when $\int(X(t) - \mu(t))$ increases—$\zeta^k$ becomes negative.

It is commonly justified to approximate Eq 2 as a finite sum

$$X(t) \approx X_n(t) = \mu(t) + \sum_k^n \zeta^k \phi^k(t). \tag{4}$$

It should be noted that $\phi^i(t)$, $i = 1, 2, \ldots$ is a basis of the functional space in $H$.

This understanding of the underlying process permits the application of fPCA. A smoothed representation with the basis functions (e.g. B-splines) fulfilling Eq 4 can be obtained using $L^2$ regularisation. To reduce the dimensionality to $K$, we keep only the first $K$ components (i.e. $\phi^i(t)$) that represent the dominant mode of variation in $X$ by setting the first component to

$$\phi^1 = \underset{\|\phi\|=1}{\arg\max}\left\{ \mathrm{Var}(\int_{\mathcal{T}}(X(t) - \mu(t))\phi(t)dt) \right\}, \tag{5}$$

and the following $K - 1$ components to

$$\phi^k = \underset{\|\phi\|=1, \langle\phi,\phi^j\rangle=0 \text{ for } j=1,\ldots,k-1}{\arg\max}\left\{ \mathrm{Var}(\int_{\mathcal{T}}(X(t) - \mu(t))\phi(t)dt) \right\}. \tag{6}$$

$\|\phi\|$ is the square norm, i.e. $\| \phi \| = \sqrt{(\int \phi(t)^2)}$. It should be emphasised that $\phi^k$ can differ by a factor of -1 due to the square norm, and consequently, the operator $\zeta^k$ (Eq 3) can be either positive or negative depending on $\phi^k$. This means that the slope of the cluster-dividing boundary can be pointing upwards or downwards and still describe the same functional composition.

We exemplified the impact of the first two fPCs to analyse the consequences on nucleosome phasing in chromatin remodeler-deficient cells (see for example Figs 1, 2 and 4). It should be noted that the fPCs were amplified to highlight their functional contribution. We set the scaling factor to $\zeta^1 = \zeta^2 = 20$ in all figures that demonstrate their effects (i.e. magenta shows the effect of the fPC multiplied by 20 and added to the mean, and green depicts the fPC multiplied by 20 and subtracted from the mean). The determined factors were predominantly distributed in $\zeta^{1, 2} \in [-20, 20]$ for all strains and replicates, and most of them were in fact much lower. Therefore, we limited the scaling of the axes for $\zeta^1$ and $\zeta^2$ to $[-20, 20]$ for all plots that show the cluster distribution with respect to the factors. Therefore, all figures and axes were directly comparable. The few outliers that were outside this range were incorporated into the analysis despite of being not shown in those plots. In most cases, only 1 or 2 genes were outside these ranges. However, large genes for *isw1Δ* replicate A and *isw2Δrsc8chd1Δ* replicate B had 71 and

72 outliers, respectively. Nonetheless, this value amounts to less than 2%/ of all considered protein-coding profiles for these setups.

In order to validate that nucleosome arrangements tend to separate along the first two fPCs when using a different clustering method, we repeated grouping using WARD with an Euclidean distance metric on the profiles. Clustering was implemented using the `AgglomerativeClustering` class of the `scikit-learn` Python package. In contrast to the Pearson correlation index, the Euclidean distance does not have an upper bound, and clustering was more sensitive to outliers. We removed 29 genes who had an absolute fPC score larger than 20 for either of the two fPCs. Despite the fact that the boundary is less neat than when using the *k*-mean clustering, our results suggest that the WARD gene groups tend to separate along the first two fPCs as well (S14 Fig).

## Quantifying the cluster boundary

Long genes were linearly separable with respect to the Pearson coefficient clusters in all WT and mutant conditions. The boundary was determined using a linear SVM. We ignored the prediction error and the intercept of the linear boundary, and instead considered only the slope differences between the two replicates. As aforementioned, the sign of the slope $m$ does not matter, and we consider therefore only $|m|$. To quantify the variability in the two replicates, we introduce the following measurement

$$s(i) = \frac{(\bar{m}_i - \bar{m}_{WT})^2}{(|m_i^A| - |m_i^B|)(|m_{WT}^A| - |m_{WT}^B|)}. \tag{7}$$

$\bar{m}$ denotes the average over the absolute slopes of both replicates. We defined a change as notable when $s(i) > 1$, which implies that the mean variability between WT and mutant is larger than the variability within the replicates, i.e.

$$(\bar{m}_i - \bar{m}_{WT})^2 > (|m_i^A| - |m_i^B|)(|m_{WT}^A| - |m_{WT}^B|). \tag{8}$$

As we consider only two replicates, we restrain from using the word *significant* as much as possible and use *noteworthy* or *notable* instead.

The slope of the boundary $m$ indicates the contribution of each fPC to describe the discriminator between the clusters. As $m$ shows the change of $\zeta^2$ over one unit of $\zeta^1$, we can determine the separating boundary by

$$\phi' = \frac{m\phi^1 + \phi^2}{m + 1}. \tag{9}$$

The impact of $\phi'$ can be visualised by multiplying a scaling factor which is followed by addition to and subtraction from the mean. In this study, we used a factor of $\zeta' = 5$ to create the grey bands in the plots that show the effect of the separating function.

## Measuring interdependence between nucleosome phasing and other nuclear properties

In order to analyse interdependence of nucleosome positioning with other nuclear properties, we divided all factors into two equally sized cluster using the median wherever possible. For example, the half with the smaller NDRs was assigned to group -1, whereas the larger half was group 1. This split was performed after filtering for the size (i.e. large or small genes). The analysis aimed to find a correlation between nuclear factor group and Pearson cluster. To remove

any bias with respect to the group size, we forced both Pearson clusters to contain the same number of genes.

We used a simple feedforward network with no hidden neurons and a single output neuron whose activation indicated the predicted Pearson cluster. The number of input neurons varied between 1 and 2, depending on whether we considered a multivariate interdependence. The group of the nuclear factor (i.e. -1 or 1) was set as input neuron activation. This was weighted and summed together with all other input values. The activation function of the output was a modified sign function, which returned 0 when negative and 1 when positive. Therefore, if the weighted sum over the input was lower than or equal to 0, the output would be 0, and 1 otherwise.

Weights were trained using a Hebbian-like learning method [41]. In order to avoid any confusion, we name Pearson cluster 0 and nuclear factor group -1 *low* cluster, whereas we define group 1 in both cases to be the *high* cluster. The weight was defined to be the average number of genes where the nuclear factor group and Pearson coefficient cluster where both *low* or both *high*; minus the average number where one of them was *low* whilst the other *high*. The implementation as a neural network allowed the straightforward extension to compare interdependence with several factors at the same time using the same method.

## Supporting information

**S1 Fig. Cluster significance test for all genes in WT replicate A.** We measure the inter-cluster correlation for the found clusters and 500 random clusters. It is expected that the found gene groups have a significantly lower inter-cluster correlation than a random separation (i.e. the found clusters are most dissimilar based on the Pearson correlation). A KS test over 500 randomly sampled inter-cluster correlation indices for the $k$-mean and random grouping, respectively, proved their significance. Magenta shows the distribution for the found clusters, orange displays one example for random clustering. (A) The distribution over the absolute pairwise inter-cluster correlations is significantly lower for the gene groups determined by the $k$-mean clustering than for the random grouping. (B) The cumulative distribution function for the found clusters raises much more quickly. A KS test verified that this trend is indeed significant (average p-value over 500 repetitions 0.0009).
(TIFF)

**S2 Fig. The WT fPC scores $\zeta$ coloured with respect to the Pearson clustering using all genes.** Blue and orange indicate each one group, the dashed line symbolises the best linear separation using a SVM. The x-axis represents the score of the first fPC $\zeta^1$, the y-axis gives the score for the second fPC $\zeta^2$. All axes are scaled to the same size; shapes are therefore comparable. (A) and (B) show all genes for replicate A and B. (C) and (D) display the fPC scores after filtering for large genes (>1000 bp) for replicates A and B. (E) and (F) show small genes (($\leq$1000 bp) for replicate A and B. We removed the separating boundary because it did not reasonably divide the clusters. Nevertheless, we kept the estimated linear function in the legend to allow a comparison with other boundaries. Of particular note is the bias, which can be even order of magnitudes different from large-gene clusters. (G) and (H) display the fPC scores after filtering for very large genes (>3000 bp) for replicates A and B.
(TIFF)

**S3 Fig. First 10 fPCs for WT A all genes.** FPCs are ordered with respect to how much variance they explain (i.e. fPC1 explains the most whereas fPC10 explains the least). The mean is given as a black dashed line, a positive contribution is shown in magenta, whereas a negative contribution is displayed in green. The first two fPCs are the ones that were presented in Fig 1

(F) (21.3% and 11.5% explained variance, respectively). The fPCs that follow after the major two ones become increasingly complex, and it is difficult to quantify their effect in a straight-forward measurement. With the exception of fPC4 (7.8% explained variance), the plots suggest that the effects of the fPCs that were not included would not have been captured by the linear correlation index, as they describe changes specific to a single nucleosome (e.g. fPC6) or complex changes (e.g. fPC7). We want to remind that the variance captured by the fPCs could either moderately occur along the majority of genes; or alternatively, there is a strong effect on a small subset of profiles. We interpret the results as follows. FPC1 and fPC2 show global trends how nucleosome arrangements change along most protein-coding regions. This is emphasised by the fact that they capture the most deviance from the mean and that their effects are not specific to particular nucleosome positions. Many other fPCs (e.g fPCs 5, 6, and 7, 7.5%, 7%, and 6.1% explained variance, respectively) include strong position-specific effects. Despite the large impact on the amplitude at precise positions (e.g. +3 or +4), the explained variance by these fPCs is lower. We presume that this indicates a strong variance at these positions in a small subset of genes. Thus, they show a much lower variance. We focus on the first two fPCs because they can separate the Pearson clusters, which in turn indicates that they capture global trends since they were determined over all genes.
(TIFF)

**S4 Fig. Heatmaps for small-gene nucleosome profiles reveal antagonistic roles for Rsc8 and Chd1 to establish phasing boundaries.** Cluster 1 and 2 for all genes in WT conditions were plotted only including small genes on the left. Indeed, correct positioning is either completely disrupted (Cluster 1), or clear phasing is lost after +3 or +4 position and individual peaks do not stand out thereafter (Cluster 2). However, both Pearson clusters for *rsc8*-depleted cells (centre) show clear phasing probabilities, despite all genes being smaller than the considered 1000 bp after the +1. The double mutant *chd1Δrsc8* seems to re-establish the gene boundaries for nucleosome phasing, as positioning is either disrupted (Cluster 2, compare with Cluster 1 in WT) or does not exhibit clearly distinguishable peaks after the +3 or +4 nucleosome (Cluster 1, compare with Cluster 2 in WT). Defining a group as being 1 or 2 was arbitrary and has no significance. Copper values show large MNase-seq signal values, whereas dark segments indicate a low amplitude. Values in between are uniformly scaled.
(TIFF)

**S5 Fig. The large-gene fPC effect in WT.** Despite fact that the functions differ in the A and B replicate ((A) and (B)), they both describe the same properties as when considering all genes (Fig 1(F)). To be precise, the first fPC describes seemingly position-dependent scaling (grey vertical bars), and the second explains coordinated phasing (grey arrows). The mean is displayed as a black dashed line, whereas a positive and a negative functional contribution are given in magenta and green, respectively.
(TIFF)

**S6 Fig. The Pearson coefficient clusters for exclusively small genes correspond to the gene size.** When we repeated the Pearson coefficient clustering considering exclusively small genes, we can linearly separate again the two groups (orange and blue). However, this is predominantly explained by the size of the gene (short pink, long green). This in line with the hypothesis that coordinated nucleosome phasing along the transcribed region is strictly limited within the gene body. The phase separating line was determined on the Pearson clusters (dashed black line) using an SVM. The same separating boundary was also plotted in right plot showing grouping with respect to the size. We plotted the original SVM boundary from the Pearson clusters with a dashed grey line to indicate that it was not determined using gene size. (A) and

(B) give the Pearson clusters for replicate A and B. (C) and (D) show the size dependence of replicate A and B.
(TIFF)

**S7 Fig. Pearson clusters of small genes lose separability with respect to their fPC scores.** The figure shows the fPC scores $\zeta$ of small genes ($<$1000 bp) of all conditions coloured with respect to the *all-gene* Pearson clustering. Blue and orange indicate each one group, the dashed line symbolises the best linear separation using a SVM. We removed the linear boundary in plots where it went through the periphery instead of dividing the data points. The x-axis represents the score of the first fPC $\zeta^1$, the y-axis gives the score for the second fPC $\zeta^2$. All axes are scaled to the same size; shapes are therefore comparable.
(TIFF)

**S8 Fig. The small-gene fPC effect in *chd1Δrsc8* strains.** The double mutant seemingly re-establishes gene boundaries, and coordinated phasing is at least weakened after the +2 nucleosome (+1 in turquoise, +4 in blue, +6 in orange). This is true despite the fact that the A and B replicate differ. Figs (A) and (B) show the clusters for replicate *A* and *B*, and Figs (C) and (D) display their fPCs. We removed the separating boundaries in (A) and (B) because they did not reasonably divide the clusters. Nevertheless, we kept the estimated linear function in the legend to allow a comparison with other boundaries. Of particular note is the bias, which differs largely from large-gene clusters. The dashed black lines, the solid purple, and the solid green lines indicate the mean, a positive contribution, and a negative contribution, respectively.
(TIFF)

**S9 Fig. Cluster significance test for large genes.** We compare the inter-cluster correlation between the Pearson clusters (magenta) and random partitioning (orange). We define the inter-cluster correlation as the distribution over all absolute pairwise Pearson indices between two genes from different clusters. It is expected that the inter-cluster correlation for the grouping determined by the *k*-mean tends to be significantly lower than the random clustering. This is verified by a KS test over 500 randomly sampled inter-cluster indices for the random and *k*-mean grouping, respectively. As the sub-sampled Pearson indices for both approaches can be sometimes fairly similar, we repeated the significance test over 500 random clusters and determined the average p-value. We show here the result for one example of the 500 random groupings. Left: The inter-cluster correlation distribution for the determined gene groups tends to be lower than the random clustering for all strains. Right: This notion is confirmed by the cumulative distribution function (CDF), where the Pearson CDF increases more quickly than the one for random clustering.
(TIFF)

**S10 Fig. Pearson clusters of large genes are linearly separable with respect to their fPC scores (replicate A).** The figure shows the fPC scores $\zeta$ of all conditions coloured with respect to the Pearson clustering using only large genes ($\geq$1000 bp). Blue and orange indicate each one group, the dashed line symbolises the best linear separation using a SVM. The x-axis represents the score of the first fPC $\zeta^1$, the y-axis gives the score for the second fPC $\zeta^2$. All axes are scaled to the same size; shapes are therefore comparable. It should be emphasised that only the absolute slope value matters and not the sign (i.e. pointing upwards or downwards).
(TIFF)

**S11 Fig. Pearson clusters of large genes are linearly separable with respect to their fPC scores (replicate B).** The figure shows the fPC scores $\zeta$ of all conditions coloured with

respect to the Pearson clustering using only large genes ($\geq$1000 bp). Blue and orange indicate each one group, the dashed line symbolises the best linear separation using a SVM. The x-axis represents the score of the first fPC $\zeta^1$, the y-axis gives the score for the second fPC $\zeta^2$. All axes are scaled to the same size; shapes are therefore comparable. It should be emphasised that only the absolute slope value matters and not the sign (i.e. pointing upwards or downwards).
(TIFF)

**S12 Fig. Example of an arbitrary mutant that only influences the MNase-seq average but not the collective behaviour of the nucleosome array.** (A) We consider an arbitrary but fictional chromatin remodeler mutant that causes the depletion of the +2 nucleosome. All other nucleosomes remain unperturbed and keep their positioning. The MNase-seq profile at the +2 position consists only of random noise for all genes. (B) This has visibly a strong impact on the average distribution, as the +2 position is depleted in the arbitrary mutant (blue) in comparison with the WT (bold red line). (C) Since the descriptive variance decreased in our arbitrary mutant (i.e. variation that is not attributed to random noise), the fPC scores are slightly affected. Overall, however, the boundary slope remains fairly similar to the real WT A from which we constructed the arbitrary mutant. In fact, difference to WT A is smaller than the difference between the two biological replicates. In order to make sure we consider only mutants that affect the entire nucleosome array, we selected only strains for which the boundary slope notably changed with respect to the WT.
(TIFF)

**S13 Fig. Interdependence of Pearson clusters of MNase-seq profiles and other nuclear factors.** The orange bar shows the ratio of cases where the nuclear factor could predict clustering, blue gives the wrongly classified ratio. Random guessing would be correct in 50% of the cases, which is given by the dashed black line. Consequently, the orange bar must exceed the dashed line to suggest interdependence.
(TIFF)

**S14 Fig. Clustering using WARD and an Euclidean distance metric indicates also a separation along the first two fPCs.** We repeated gene grouping (WT A, large genes) with WARD as a fundamentally different clustering method on the nucleosome profiles using an Euclidean distance metric. However, contrary to the Pearson correlation, the Euclidean distance does not have an upper bound, and clustering was more sensitive to outliers. We removed 29 genes who had an absolute fPC score larger than 20 for either of the two fPCs. (A) The silhouette criterion indicates once again that when dividing genes into clusters, it is best to separate them into two groups. (B) Despite the fact that the Euclidean distance measures a different property than the Pearson correlation and although WARD functions fundamentally differently, the gene groups tend to separate into two groups along the first two fPCs. The boundary is admittedly not as neat as for the $k$-mean clusters in WT A, but they are comparable to other mutants in the study. It is expected that the boundary itself changes, as the Euclidean metric captures different properties. We can conclude that the separation of nucleosome profiles into different gene groups along the major two fPCs is not merely an artifact of our methodology.
(TIFF)

**S15 Fig. Clustering distribution using PCA and their principal components.** Indeed, conventional PCA can separate the clusters for all genes in WT conditions ((A) and (B) for replicate *A* and *B*) similarly to fPCA. The two clusters are given in blue and orange. However, the two determined PCs ((C) and (D) for replicate *A* and *B*) differ slightly with respect to the fPCA

due to the independence assumption. Here, light blue and light orange indicate PC1 and PC2, respectively.
(TIFF)

**S16 Fig. AT-ratio distribution with respect to the fPC scores.** Whilst there is seemingly a slight correlation between Pearson coefficient clusters and AT-ratio in the *A* replicate, this is trend vanishes for the *B* replicate. In fact, both replicates might rather distribute AT-rich and AT-poor genes orthogonal to the dividing boundary. We plotted the original SVM boundary from the Pearson clusters with a dashed grey line to indicate that it was not determined using the AT content.
(TIFF)

## Author Contributions

**Conceptualization:** Leo Zeitler, Cyril Denby Wilkes, Julie Soutourina, Arach Goldar.

**Data curation:** Leo Zeitler, Kévin André, Cyril Denby Wilkes, Arach Goldar.

**Formal analysis:** Leo Zeitler, Arach Goldar.

**Funding acquisition:** Julie Soutourina.

**Investigation:** Leo Zeitler, Kévin André, Cyril Denby Wilkes, Julie Soutourina, Arach Goldar.

**Methodology:** Leo Zeitler, Cyril Denby Wilkes, Arach Goldar.

**Project administration:** Julie Soutourina.

**Supervision:** Cyril Denby Wilkes, Julie Soutourina, Arach Goldar.

**Validation:** Leo Zeitler, Adriana Alberti, Julie Soutourina, Arach Goldar.

**Visualization:** Leo Zeitler, Arach Goldar.

**Writing – original draft:** Leo Zeitler, Adriana Alberti, Cyril Denby Wilkes, Julie Soutourina, Arach Goldar.

**Writing – review & editing:** Leo Zeitler, Adriana Alberti, Cyril Denby Wilkes, Julie Soutourina, Arach Goldar.

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
