## [Decision Letter · Decision Letter 0]

19 Aug 2023

Dear Dr Goldar,

Thank you very much for submitting your manuscript "A Genome-Wide Comprehensive Analysis of Nucleosome Positioning in Yeast" for consideration at PLOS Computational Biology.

As with all papers reviewed by the journal, your manuscript was reviewed by members of the editorial board and by several independent reviewers. In light of the reviews (below this email), we would like to invite the resubmission of a significantly-revised version that takes into account the reviewers' comments.

The reviewers have identified numerous issues that should be addressed in your revision. Two of the reviewers also noted that relevant code was not made available for review via Github or similar public repositories. Please address this in your revision. 

We cannot make any decision about publication until we have seen the revised manuscript and your response to the reviewers' comments. Your revised manuscript is also likely to be sent to reviewers for further evaluation.

Sincerely,

Shaun Mahony

Academic Editor

PLOS Computational Biology

Sushmita Roy

Section Editor

PLOS Computational Biology

Reviewer's Responses to Questions

**Comments to the Authors:**

Reviewer #1: Summary

In this work, Zei et al. employ Functional Principal Component Analysis (fPCA) to analyze nucleosomal profiles in S. cerevisiae, treating nucleosomal array patterns within coding regions as single units and simultaneously split the two properties (phasing and amplitude) in nucleosomal arrays. Using fPCA, they uncover Rsc8's influence in phasing nucleosome positions in small genes, and Chd1's role after the +1 nucleosomes, shedding light on their antagonistic effect on nucleosome phasing both locally and over long distances. Although fPCA is well-established, this novel use reveals unique insights into chromatin remodeling mechanisms but suggests areas in their analytical framework that could benefit from further refinement.

Major Concerns

1. Since the analysis mainly relies on Pearson coefficients, please include a brief description of their measurement and clustering in the main text and detailed information in the Method section. I recommend merging Fig 7 and Fig 1 for a comprehensive illustration of the process.

2. Clarify the rationale for clustering Pearson coefficients and how it aligns with the interpretation of the two Pearson clusters.

3. What is the justification for using 20 B-splines, and how does it compare to conventional smoothing techniques?

4. Could the framework's effectiveness be compromised if the MNase-seq signal doesn't align with a wave-like distribution (lines 129-132)?

5. Explain the choice of the first two functional Principal Components (fPCs) for nucleosome profile analysis (line 138). What criteria were used, and will the third fPC provide additional insights?

6. In result2, disclose the ratio between small and large genes. Does the majority of large genes affect the similarity with all-gene fPCs?

7. In line 155, elaborate on the size-dependent phasing of nucleosome positions with rsc8-depleted strain. Are the results consistent with the mutation data in the authors' recent publication?

8. Consider shifting SFig A6 to the main figure, possibly merging it with Fig 2 to clarify Chd1 and Rsc8 roles.

9. In line 204, detail how gene size-dependent bias removal was confirmed and discuss how other potential biases were addressed in key factor comparisons.

10. Explain the removal of isw1Drsc8D from the analysis in lines 207-208. What makes them exceptions?

11. In line 313, clarify the differences between replicate A and B, as they may be due to the nonlinear effect of double mutants, not just "experimental variability."

12. Regarding lines 234-236, clarify the results leading to the conclusion about notable alterations in coordinated phasing. Please elaborate on why the focus is on Chd1 and its effect after the +1 nucleosomes.

13. In the last result, consider presenting a correlation or causal relationship between remodeler deletions and genomic properties, perhaps using Bayesian network or structural equation modeling, if possible.

Minor Concerns

1. Verify the reference in line 181 to "SFigs A.2(G, H)," as it seems to be missing.

Reviewer #2: The question of how different remodelers move nucleosomes is of fundamental importance and has not been completely understood. Here the authors have performed a reanalysis of the experimental dataset of Ocampo et al using a new computational method of functional PCA. I think, this is an interesting method, which should work fine theoretically. I think it is important to prove that this method is reproducing the results obtained with other methods like classical PCA or other clustering which has been applied to MNase-seq analysis previously. On the other hand, if the results are different, then it would be good to explain what’s different, why, and what’s the biological meaning of the clusters that are uniquely identified by fPCA and cannot be found by other clustering methods. More detailed suggestions are listed below in the order of the manuscript flow.

Lines 60-62: “It has been suggested that high gene expression correlates with low nucleosome regularity, which is characterised by weak phasing and extreme spacing (both short and long) [18, 28].” I think this statement contradicts the conclusions of Ref 31 which shows stronger phasing at highly expressed genes. Refs 18 and 28 cited here probably also do not support this statement in this form.

Lines 78-81: “To our knowledge, a single mathematical framework assessing gene-wide nucleosome phasing has not been proposed, and a direct comparison of the effects in different remodeler-deficient strains is difficult”. I am not sure what is expected as a "single mathematical framework", but a number of publications, including e.g. Ref 31 cited here, developed different frameworks for this purpose.

Lines 83-85: “By doing so, we address two points that have previously largely fallen short. Firstly, we evaluate long-range influences of positioning within transcribed regions that exceed neighbouring nucleosomes”. Autocorrelation-based approaches also consider distances between nucleosomes beyond next neighbours, so this statement needs to be refined to explain the difference of the current approach a bit better.

Lines 129-130: “..the link between linear correlation and coordinated 129

nucleosome phasing.” Linear correlation of what?

Figure 1: I suggest to add the percentage of explained variance to each principal component on the axes. Also, please rewrite the legend to explain what exactly is schematically drawn or plotted in each panel.

Line 144: “Astonishingly…” Unclear why this word is used.

Figure 2: Panels A-C here seem to report the main finding of the disappearance/reappearance of the cluster separation. Please try to reproduce the effect of disappearance/reappearance of the cluster separation using some other method (e.g. perform k-means clustering with one of standard software packages that exist for classical MNase-seq analysis), or explain why it can not be reproduced using another method. If it can not be reproduced with other methods it is really important to explain what these clusters represent, both mathematically and biologically. Are they characterised by different averaged nucleosome profiles? Then it would be good to show such different averaged profiles in the manuscript.

Line 207: The authors have removed some data as outliers. It would be good to list the total number of datasets before/after the removal of this outlier dataset.

Figures 3, 4, 5: It would be good to rewrite the figure legends to detail what is plotted on each panel. Also, it would be better to position figure labels (A), (B), (C), (D) above the corresponding panel, not below.

Figure 5: It would be good to rescale the Y axis to show the whole bar. Starting the bar from 0.4 is a bit misleading.

Is Figure 5 showing model performance around 50% (60% in best cases)? I am a bit confused if it is so. A performance of a ML model with ~50% correct predictions means that the model's performance is comparable to random guessing, so this aspect needs to be a bit better explained.

Methods section: in general, it needs to be upgraded to make the description self-sufficient to reproduce. More specific points about the Methods section are listed below:

Please describe in more detail all parameters, and if the calculation was done with in-house scripts, please include these scripts as supplementary materials or make them available online and provide a link.

Line 458: “…inter and intra Pearson correlation…”. Unclear what this means.

Please explain all abbreviations, e.g. “JS”, “PI”, etc.

Figure 7: several points about this figure are listed below:

It's unclear what Figure 7A means. Please add proper captions on the X and Y axes, move the color-code legend away from the Y-axis and rescale it to avoid impression that it's related to the Y-axis caption and add more detailed description to the figure legend.

I suggest to make heatmaps as in Figures 7C and 7D, separately for small and for large genes, and include them in the main manuscript to support the main conclusions of the study.

I suggest to indicate on the Y axis of each heatmap figure, how many genes is in each cluster. Please move the color-code legend away from the Y axis, otherwise it can be confused with the numbers of genes (the numbers of horizontal lines on the heatmap)

“A decrease of nucleosome as a function of distance”. Should it be “A

decrease of nucleosome phasing as a function of distance”?

Lines 515-516: “As we consider only two replicates, we restrain from using the word significant as much as possible and use noteworthy or notable instead.” It would be still good to explain in the manuscript, which results are statistically significant and which are not.

Reviewer #3: Zeitler et al. describe a novel method of comparing MNase-seq datasets in an unbiased manner. Using fPCA of MNase composite plots for every gene, they characterized global trends in MNase sensitivity in a variety of previously published MNase-seq datasets performed on chromatin remodeler-deletion yeast strains. Their primary conclusions include establishing the RSC complex as required for global nucleosome positioning and showing evidence for Chd1 being primarily responsible for local arrangements of nucleosome positioning at the start of gene bodies. The technique demonstrated (fPCA) is novel for this application and appears to work at a global level. However, the largest outstanding concern is not with the nature of the bioinformatic analysis itself, but at a biochemical level regarding whether the distinct MNase datasets are comparable at the global level.

Major Points:

• This paper reanalyzes MNase- seq data generated from two distinct publications (Ocampo 2019 and Ocampo 2016). However, it is unclear from the analysis performed within the paper whether the MNase datasets were fundamentally comparable in the context of this analysis. Slight technical changes in MNase concentration have previously been demonstrated to dramatically change the relative occupancy of nucleosomes across the gene body and changes in MNase digestion paired to specific size selection can enrich for variably phased nucleosomes (https://pubmed.ncbi.nlm.nih.gov/22559821/). It would be critical to demonstrate the datasets are comparable from an MNase digestion perspective. More generally, are there genomic regions across the deletion strains that do not change in nucleosome phasing or occupancy?

• It is unclear what the implications of this statement mean for their algorithm. Are these false-negative datasets or does this speak to variability in MNase digestion profile?

"For a correct interpretation of the results, it is crucial to highlight that this does not imply that other mutants had no effect on the nucleosome profile. In fact, other gene deletions that were not marked as being significant resulted also in visibly altered MNase-seq signals."

Minor Points:

• Legend in Figure 1B, 2A, 2B, 2C, 3-all, 5B, 5D, 5F are obscuring data and full interpretation

• Same issue with scatterplots in supplementary figures

**Have the authors made all data and (if applicable) computational code underlying the findings in their manuscript fully available?**

Reviewer #1: **No: **The authors indicated that all codes would be available on GitHub, but I was unable to locate the link.

Reviewer #2: **No: **The manuscript reports a computer code which can be made available

Reviewer #3: **No: **Cover letter notes state code will be made available on Github, but no link was found in the manuscript making evaluation of the exact implementation of their equations impossible.

PLOS authors have the option to publish the peer review history of their article (what does this mean?). If published, this will include your full peer review and any attached files.

Reviewer #1: No

Reviewer #2: No

Reviewer #3: No
---

## [Decision Letter · Decision Letter 1]

16 Oct 2023

Dear Dr Goldar,

Thank you very much for submitting your manuscript "A Genome-Wide Comprehensive Analysis of Nucleosome Positioning in Yeast" for consideration at PLOS Computational Biology.

As with all papers reviewed by the journal, your manuscript was reviewed by members of the editorial board and by several independent reviewers. In light of the reviews (below this email), we would like to invite the resubmission of a significantly-revised version that takes into account the reviewers' comments.

As you will see, Reviewers 1 and 2 found that you addressed several of the points that they brought up in their initial reviews, although both point out several relatively minor items that were not sufficiently addressed. Reviewer 3 points to a lack of attention to several important issues that they brought up in their initial review. In particular, it will be important to address their question about the false negative rate of your approach. I also agree with the reviewer that it should be straightforward to edit the scatter plots in your manuscript such that the legends do not obscure the data. Unfortunately, we will be unable to consider publication of the manuscript until all reviewer comments have been sufficiently addressed.

We cannot make any decision about publication until we have seen the revised manuscript and your response to the reviewers' comments. Your revised manuscript is also likely to be sent to reviewers for further evaluation.

Sincerely,

Shaun Mahony

Academic Editor

PLOS Computational Biology

Sushmita Roy

Section Editor

PLOS Computational Biology

As you will see, Reviewers 1 and 2 found that you addressed several of the points that they brought up in their initial reviews, although both point out several relatively minor items that were not sufficiently addressed. Reviewer 3 points to a lack of attention to several important issues that they brought up in their initial review. In particular, it will be important to address their question about the false negative rate of your approach. I also agree with the reviewer that it should be straightforward to edit the scatter plots in your manuscript such that the legends do not obscure the data. Unfortunately, we will be unable to consider publication of the manuscript until all reviewer comments have been sufficiently addressed.

Reviewer's Responses to Questions

**Comments to the Authors:**

Reviewer #1: Regarding the first point about explained variance in the principal component analysis on Page 17/115, the authors chose not to include the percentage of variance explained, reasoning that it's irrelevant to their analysis. While I understand the argument, I still believe a supplementary plot resembling Figure 1(F) would be beneficial. This plot should illustrate the variance explained by the top 10 functional Principal Components (fPCs), followed by a discussion on why the values are low. The lower explained variance might indicate limitations in the model's ability to capture the complexity of the data.

Concerning the methodology on Pages 5/115 and 17/115, the authors have not yet fully addressed questions regarding the suitability of Pearson correlation and k-means clustering. To enhance the rigor of the study, it would be instructive for the authors to explore alternative correlation methods and clustering algorithms to verify that their findings are not artifacts of the selected methodology. If Pearson correlation and k-means clustering are particularly well-suited to this problem, a detailed justification would be appreciated.

For Figure 1B and C, it would be helpful if the authors could clarify the choice of the color scale for the heatmaps. Is the chosen color scale perceptually uniform? On Line 153, the phrase “a different description of it” is a bit vague; some clarification would be beneficial. Similarly, at Line 207, given that a majority (~73%) of all genes in the study are large, the authors should discuss the potential bias in the findings regarding gene similarity.

With respect to Figure 2, the authors mention in the legend for panels D and F that the wave-like pattern in the second functional Principal Component (fPC) dissipates after the +2 nucleosomes. However, this doesn't seem to align with the data presented. I would like the authors to clarify this discrepancy. Also, it would improve readability if the legend information for panels A-C and D-F were grouped together.

Lastly, the issue regarding the isw1 deletion mutant deserves more attention. The mutant potentially affects both Isw1a and Isw1b complexes, which have distinct functionalities according to Yen et al., 2012. I would like the authors to consider whether the observed effects from +2 nucleosomes onwards could be attributed to Isw1b and provide their thoughts on this matter.

Reviewer #2: The authors have addressed most of my points, the method is well explained and the manuscript is now much clearer. The only point from my initial review that still needs a bit of attention is this one:

"Figure 2: Panels A-C here seem to report the main finding of the disappearance/reappearance of the cluster separation. Please try to reproduce the effect of disappearance/reappearance of the cluster separation using some other method (e.g. perform k-means clustering with one of standard software packages that exist for classical MNase-seq analysis), or explain why it can not be reproduced using another method. If it can not be reproduced with other methods it is really important to explain what these clusters represent, both mathematically and biologically. Are they characterised by different averaged nucleosome profiles? Then it would be good to show such different averaged profiles in the manuscript."

To clarify, I suggest a couple of very small action points here:

1) The authors can at least mention NucTools that offers a similar functionality of k-means clustering of MNase-seq (https://bmcgenomics.biomedcentral.com/articles/10.1186/s12864-017-3580-2).

2) It would be still good to clarify, what is the biological and mathematical meaning of these clusters. I suggested initially to show the averaged nucleosome profiles for each cluster. I meant the aggregate profiles of the nucleosome occupancy for each cluster (not the principal components, but the actual nucleosome occupancy), and then discuss briefly the biological meaning of these profiles. Some hint to such biological interpretation is already provided in the schematic figure 1D, and it would be good to support it by actual nucleosome occupancy profiles.

Reviewer #3: The authors have responded to my concern regarding MNase digestion bias. However, they notably have declined to address any of the other issues.

My original major concern still stands which is: what is the false-negative interpretation rate of their approach? The authors modified the text I referenced in the original manuscript, but the replacement text is equally unclear. If the authors find visual differences in the chromatin remodeler deletion strains but their fPCA implementation does not identify them as significant, is this not a critical issue in their approach? What is meant by a 'gene-specific' variance and why would their approach not identify this?

Finally, we want to re-emphasize that purposefully providing occluded figures throughout the manuscript is not 'a shame' as the authors mention, but bad science. There is no justification for not moving figure legends outside the chart area

**Have the authors made all data and (if applicable) computational code underlying the findings in their manuscript fully available?**

Reviewer #1: Yes

Reviewer #2: Yes

Reviewer #3: Yes

PLOS authors have the option to publish the peer review history of their article (what does this mean?). If published, this will include your full peer review and any attached files.

Reviewer #1: No

Reviewer #2: No

Reviewer #3: No
---

## [Decision Letter · Decision Letter 2]

18 Dec 2023

Dear Dr Goldar,

Thank you very much for submitting your manuscript "A Genome-Wide Comprehensive Analysis of Nucleosome Positioning in Yeast" for consideration at PLOS Computational Biology. As with all papers reviewed by the journal, your manuscript was reviewed by members of the editorial board and by several independent reviewers. The reviewers appreciated the attention to an important topic. Based on the reviews, we are likely to accept this manuscript for publication, providing that you modify the manuscript according to the review recommendations.

Please see the remaining minor comments from Reviewer 1. We expect that these comments will be straightforward to deal with by making minor adjustments to the manuscript text. After these adjustments have been made, we will editorially evaluate whether you have addressed the reviewer's comments. We do not anticipate that we will require another round of reviews at this point, and we will hopefully proceed swiftly to accepting the manuscript. 

Sincerely,

Shaun Mahony

Academic Editor

PLOS Computational Biology

Sushmita Roy

Section Editor

PLOS Computational Biology

Please

Reviewer's Responses to Questions

**Comments to the Authors:**

Reviewer #1: - Regarding the first point about explained variance in the principal component analysis on Page 17/115, the authors chose not to include the percentage of variance explained, reasoning that it's irrelevant to their analysis. While I understand the argument, I still believe a supplementary plot resembling Figure 1(F) would be beneficial. This plot should illustrate the variance explained by the top 10 functional Principal Components (fPCs), followed by a discussion on why the values are low. The lower explained variance might indicate limitations in the model's ability to capture the complexity of the data.

This reviewer is satisfied with the response.

- Concerning the methodology on Pages 5/115 and 17/115, the authors have not yet fully addressed questions regarding the suitability of Pearson correlation and k-means clustering. To enhance the rigor of the study, it would be instructive for the authors to explore alternative correlation methods and clustering algorithms to verify that their findings are not artifacts of the selected methodology. If Pearson correlation and k-means clustering are particularly well-suited to this problem, a detailed justification would be appreciated.

This reviewer is satisfied with the response.

- For Figure 1B and C, it would be helpful if the authors could clarify the choice of the color scale for the heatmaps. Is the chosen color scale perceptually uniform? On Line 153, the phrase “a different description of it” is a bit vague; some clarification would be beneficial. Similarly, at Line 207, given that a majority (~73%) of all genes in the study are large, the authors should discuss the potential bias in the findings regarding gene similarity.

The response clarifies the normalization of the heatmaps, with the largest values displayed in the strongest copper hue and the lowest values in black. However, the explanation does not address whether the color scale is perceptually uniform. Perceptual uniformity is crucial for accurately reflecting differences in the data through visual representation.

In their analysis, the authors exclusively focus on large genes, which constitute the majority of the protein-coding regions studied. This selection might introduce bias, especially if conclusions drawn from large genes are generalized to the entire genome. The latter includes shorter genes, which, while they comprise only about 27% of the genome, are nonetheless significant. To enhance the robustness of their study, the authors should discuss how this focus on large genes may affect the overall applicability of their results. Moreover, they should outline any measures taken to mitigate potential bias or clarify that their conclusions are applicable primarily to large genes.

- With respect to Figure 2, the authors mention in the legend for panels D and F that the wave-like pattern in the second functional Principal Component (fPC) dissipates after the +2 nucleosomes. However, this doesn't seem to align with the data presented. I would like the authors to clarify this discrepancy. Also, it would improve readability if the legend information for panels A-C and D-F were grouped together.

This reviewer is satisfied with the response.

- Lastly, the issue regarding the isw1 deletion mutant deserves more attention. The mutant potentially affects both Isw1a and Isw1b complexes, which have distinct functionalities according to Yen et al., 2012. I would like the authors to consider whether the observed effects from +2 nucleosomes onwards could be attributed to Isw1b and provide their thoughts on this matter.

In light of the feedback regarding the isw1∆ deletion mutant, it appears that the distinction between the functionalities of the Isw1a and Isw1b complexes has not been explored in this study. Given that these complexes have distinct roles as reported in the literature, it would be valuable for the readers to understand that the observed effects from the +2 nucleosomes onwards may not be clearly attributable to one specific complex. We recommend that the manuscript explicitly acknowledge this limitation.

Reviewer #2: The authors have addressed my comments.

Reviewer #3: Authors have sufficiently responded to my concerns

**Have the authors made all data and (if applicable) computational code underlying the findings in their manuscript fully available?**

Reviewer #1: Yes

Reviewer #2: None

Reviewer #3: Yes

PLOS authors have the option to publish the peer review history of their article (what does this mean?). If published, this will include your full peer review and any attached files.

Reviewer #1: No

Reviewer #2: No

Reviewer #3: No

Figure Files:

Data Requirements:

Reproducibility:

References:

---

## [Editor Report · Decision Letter 3]

3 Jan 2024

Dear Dr Goldar,

We are pleased to inform you that your manuscript 'A Genome-Wide Comprehensive Analysis of Nucleosome Positioning in Yeast' has been provisionally accepted for publication in PLOS Computational Biology.

Best regards,

Shaun Mahony

Academic Editor

PLOS Computational Biology

Sushmita Roy

Section Editor

PLOS Computational Biology

---

## [Editor Report · Acceptance letter]

19 Jan 2024

PCOMPBIOL-D-23-01024R3 

A Genome-Wide Comprehensive Analysis of Nucleosome Positioning in Yeast

Dear Dr Goldar,

I am pleased to inform you that your manuscript has been formally accepted for publication in PLOS Computational Biology. Your manuscript is now with our production department and you will be notified of the publication date in due course.

With kind regards,

Anita Estes
